# GRAPH HOMOPHILY BOOSTER: REIMAGINING THE ROLE OF DISCRETE FEATURES IN HETEROPHILIC GRAPH LEARNING

**Ruizhong Qiu,**\* **Ting-Wei Li,**\* **Gaotang Li, Hanghang Tong**
University of Illinois Urbana–Champaign, IL, USA
{rq5,twli,gaotang3,htong}@illinois.edu

## ABSTRACT

Graph neural networks (GNNs) have emerged as a powerful approach to modeling graph-structured data and demonstrated remarkable success in many real-world applications such as complex biological network analysis, neuroscientific analysis, and social network analysis. However, existing GNNs often struggle with heterophilic graphs, where connected nodes tend to have dissimilar features or labels. While numerous methods have been proposed to address this challenge, they primarily focus on architectural designs without directly targeting the root cause of the heterophily problem. These approaches still perform even worse than the simplest multi-layer perceptrons (MLPs) on challenging heterophilic datasets. For instance, our experiments show that 23 latest GNNs still fall behind the MLP on the ACTOR dataset. This critical challenge calls for an innovative approach to addressing graph heterophily beyond architectural designs. To bridge this gap, we propose and study a new and unexplored paradigm: *directly* increasing the graph homophily via a carefully designed graph transformation. In this work, we present a simple yet effective framework called GRAph homoPHIly boosTEr (GRAPHITE) to address graph heterophily. To the best of our knowledge, this work is the first method that explicitly transforms the graph to directly improve the graph homophily. Stemmed from the exact definition of homophily, our proposed GRAPHITE creates *feature nodes* to facilitate homophilic message passing between nodes that share similar features. Furthermore, we both theoretically and empirically show that our proposed GRAPHITE significantly increases the homophily of originally heterophilic graphs, with only a slight increase in the graph size. Extensive experiments on challenging datasets demonstrate that our proposed GRAPHITE significantly outperforms state-of-the-art methods on heterophilic graphs while achieving comparable accuracy with state-of-the-art methods on homophilic graphs. Furthermore, our proposed graph transformation alone can already enhance the performance of homophilic GNNs on heterophilic graphs, even though they were not originally designed for heterophilic graphs. Our code is publicly available at `https://github.com/q-rz/ICLR26-GRAPHITE`.

## 1 INTRODUCTION

Graph neural networks (GNNs) have emerged as a powerful class of models for learning on topologically structured data. Their ability to incorporate graph topology and node-level attributes has enabled them to achieve state-of-the-art performance in a wide range of applications. These include protein function prediction, where GNNs model complex biological networks (You et al., 2021; Réau et al., 2023); neuroscientific analysis, where they are used to model brain networks (Li et al., 2023a); and social network analysis, where they help uncover patterns among users (Li et al., 2023c).

A critical challenge that many GNNs are faced with is that real-world networks can exhibit heterophily, where connected nodes tend to have dissimilar features or labels. Examples include protein–protein interaction networks where different types of proteins interact (Zhu et al., 2020),

---

\*Equal contribution.

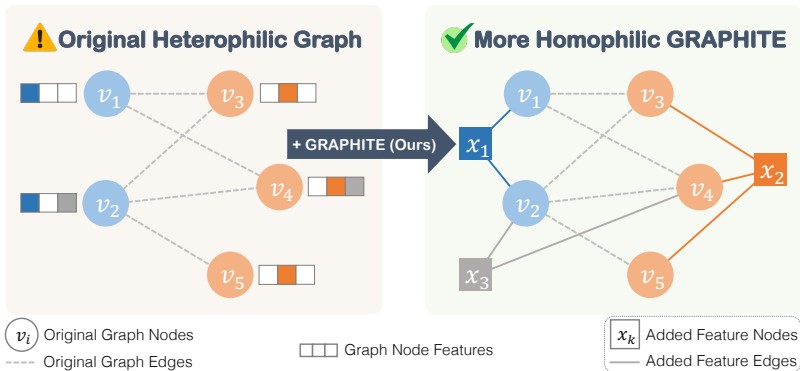

Figure 1: Illustration of our proposed GRAPHITE. Feature nodes/edges facilitate more homophilic message passing. For instance, feature node $x_1$ facilitates homophilic message passing between nodes $v_1, v_2$, and feature node $x_2$ facilitates homophilic message passing among nodes $v_3, v_4, v_5$.

or online marketplace networks where buyers connect with sellers rather than other buyers (Pandit et al., 2007). Standard GNN architectures (Kipf & Welling, 2016; Wu et al., 2019; Veličković et al., 2017; Hamilton et al., 2017; Chen et al., 2020; Abu-El-Haija et al., 2019), with their heavy reliance on neighborhood aggregation, often struggle with heterophilous graphs since aggregating features from dissimilar neighbors can dilute or distort node representations. Existing methods for heterophilic graphs mainly focus on designing new GNN architectures as workarounds for heterophilic graphs, such as separating ego and neighbor embeddings (Zhu et al., 2020; Zhang et al., 2023), incorporating multi-hop information via learnable weights (Chien et al., 2020; Dong et al., 2024), and adaptive self-gating to leverage both low- and high-frequency signals (Bo et al., 2021). More recent solutions introduce frequency-based filtering to handle both homophily and heterophily or leverage adaptive residual connections to further enhance flexibility (Xu et al., 2023; 2024a; Yan et al., 2024).

Despite plenty of architectural advances, many GNNs still perform even worse than the simplest multi-layer perceptrons (MLPs) on challenging heterophilic graphs. For instance, Table 2 shows that 23 latest GNNs still fall behind the MLP on the ACTOR dataset. This critical challenge calls for an innovative approach to addressing graph heterophily beyond architectural designs.

To bridge this gap, we propose and study a new and unexplored paradigm: *directly* increasing the graph homophily via a carefully designed graph transformation. In this work, we present a simple yet effective framework called *GRAph homoPHIly boosTEr* (GRAPHITE) to address graph heterophily. To the best of our knowledge, this work is the first method that explicitly transforms the graph to directly improve the graph homophily.

Our key idea is rooted in the exact definition of homophily/heterophily. In a homophilic/heterophilic graph, nodes that share similar features are more/less likely to be adjacent, respectively. Therefore, a natural idea to increase the graph homophily is to create "shortcut" connections between nodes with similar features so as to facilitate homophilic message passing. However, naïvely adding mutual connections between such node pairs can drastically increase the number of edges. To reduce the number of "shortcut" edges, we propose to connect such node pairs *indirectly* instead. In particular, we introduce *feature nodes* as "hubs" and connect graph nodes to their corresponding feature nodes. We further theoretically show that our proposed method can provably enhance the homophily of originally heterophilic graphs without increasing the graph size much.

Our main contributions are summarized as follows:

- **New paradigm.** We propose and study a new and unexplored paradigm: *provably* increasing the graph homophily directly via non-learning-based graph transformation. This paper is the first work on this paradigm to the best of our knowledge.

- **Proposed method.** We propose a simple yet effective method called GRAPHITE, which creates feature nodes as "shortcuts" to facilitate homophilic message passing between nodes with similar features.

- **Theoretical guarantees.** We theoretically show that GRAPHITE can *provably* enhance the homophily of originally heterophilic graphs with only a *slight* increase in size.

- **Empirical performance.** Extensive experiments on challenging datasets demonstrate the effectiveness of our proposed GRAPHITE. GRAPHITE *significantly* outperforms state-of-the-art methods on heterophilic graphs while achieving *comparable* accuracy with state-of-the-art methods on homophilic graphs. Furthermore, our proposed graph transformation alone can already enhance the performance of homophilic GNNs on heterophilic graphs.

## 2 PRELIMINARIES

### 2.1 NOTATION

An undirected graph with discrete node features can be represented as a triple $\mathcal{G} = (\mathcal{V}, \mathcal{E}, \boldsymbol{X})$, where $\mathcal{V} = \{v_1, \ldots, v_{|\mathcal{V}|}\}$ denotes the node set, $\mathcal{E} \subseteq \mathcal{V} \times \mathcal{V}$ denotes the edge set, $\boldsymbol{X} \in \{0,1\}^{\mathcal{V} \times \mathcal{X}}$ is a binary node feature matrix representing discrete node features, and $\mathcal{X} = \{1, \ldots, |\mathcal{X}|\}$ is the feature set containing all the discrete node features. In addition to that, each graph node $v_i \in \mathcal{V}$ has a node label $y_{v_i} \in \mathcal{Y}$, where $\mathcal{Y}$ is the label set with $C = |\mathcal{Y}|$ classes.

### 2.2 PROBLEM DEFINITION

In this paper, we study two key problems: (i) how to transform a graph to increase its homophily and (ii) how to perform node classification on a heterophilic graph datasets. Formally, we introduce the problem definitions as follows.

**Problem 1** (boosting graph homophily)**.** *Given a highly heterophilic graph, transform the graph to increase its homophily.* **Input:** *a heterophilic graph $\mathcal{G}$.* **Output:** *a transformed graph $\mathcal{G}^*$ with higher homophily.*

**Problem 2** (semi-supervised node classification on a heterophilic graph)**.** *Given a heterophilic graph and a set of labelled nodes, train a model to predict the labels of unlabelled nodes.* **Input:** *(i) a heterophilic graph $\mathcal{G} = (\mathcal{V}, \mathcal{E}, \boldsymbol{X})$; (ii) a labelled node set $\mathcal{V}_L \subset \mathcal{V}$ whose node labels $[y_{v_i}]_{v_i \in \mathcal{V}_L}$ are available.* **Output:** *the predicted labels of unlabeled nodes $\mathcal{V} \setminus \mathcal{V}_L$.*

## 3 PROPOSED METHOD: GRAPHITE

In this section, we propose a simple yet effective graph transformation method called *GRAph homoPHIly boosTEr* (GRAPHITE) that can efficiently increase the homophily of a graph. In Section 3.1, we will introduce the motivation of our proposed GRAPHITE. First, we will present the design of our proposed method GRAPHITE. Then, we will describe the neural architecture of our proposed method. Due to the page limit, proofs of theoretical results are deferred to the appendix.

### 3.1 MOTIVATION: NAÏVE HOMOPHILY BOOSTER

Graph heterophily is a ubiquitous challenge in graph-based machine learning. On a highly heterophilic graph, many neighboring nodes exhibit dissimilar features or belong to different classes. As a result, graph heterophily limits the effectiveness of GNN message passing, as standard aggregation schemes might fail to capture meaningful patterns in heterophilic neighbors.

Existing methods for heterophilic graphs mainly focus on designing workarounds such as new architectures or learning paradigms for heterophilic graphs, including adaptive message passing, higher-order neighborhoods, or alternative propagation mechanisms that leverage both local and global graph structures.

In contrast to existing workaround methods, we propose a new method that aims to directly increase the homophily of the graph via a specially designed graph transformation. To the best of our knowledge, this work is the first method that explicitly transforms the graph to improve the homophily of the graph.

Our idea is rooted in the exact definition of homophily and heterophily. In a heterophilic graph, nodes that share similar features are more likely to be non-adjacent. However, in a homophilic

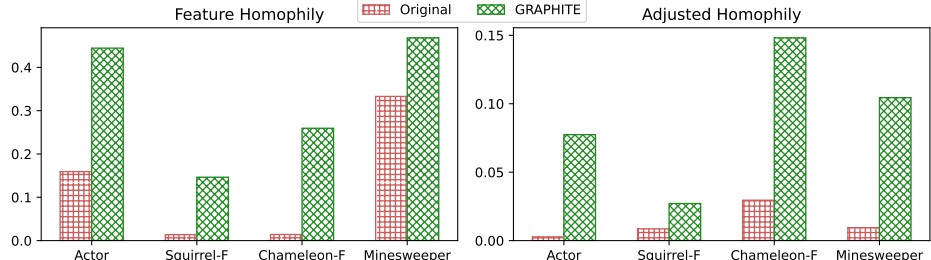

Figure 2: Our proposed GRAPHITE significantly increases the homophily of originally heterophilic graphs. We report two latest homophily metrics: *feature homophily* (Jin et al., 2022) and *adjusted homophily* (Platonov et al., 2024).

graph, nodes that share similar features should be more likely to be neighbors. Therefore, a natural idea to increase the homophily of the graph is to create "shortcut" connections between nodes with similar features, which will facilitate homophilic message passing between them.

Before we introduce the proposed method, let's consider the following naïve approach to implementing the aforementioned idea: For each pair of nodes $v_i, v_j \in \mathcal{V}$, if they share at least a feature (i.e., $\|\boldsymbol{X}[v_i,:] \wedge \boldsymbol{X}[v_j,:]\|_\infty > 0$), we add a "shortcut" edge $(v_i, v_j)$ between them. Let's call this approach the *naïve homophily booster* (NHB). The following Theorem 1 shows that NHB can indeed increase the homophily of the graph under mild and realistic assumptions.

**Theorem 1** (naïve homophily booster). *Given a heterophilic graph $\mathcal{G} = (\mathcal{V}, \mathcal{E}, \boldsymbol{X})$, let $\mathcal{E}^\dagger$ denote the set of edges after adding the NHB "shortcut" edges, and let $\mathcal{G}^\dagger := (\mathcal{V}, \mathcal{E}^\dagger, \boldsymbol{X})$ denote the graph transformed by NHB. Under mild and realistic assumptions in Appendix D.1, we have*

$$\mathrm{hom}(\mathcal{G}^\dagger) > \mathrm{hom}(\mathcal{G}), \tag{1}$$

$$|\mathcal{E}^\dagger| - |\mathcal{E}| \leq O(|\mathcal{V}|^2). \tag{2}$$

However, Equation (2) also shows that NHB is extremely inefficient despite its effectiveness in increasing homophily. For instance, even if the graph has only 2,000 nodes, NHB can add as many as 1,999,000 "shortcut" edges. The plenty of "shortcut" edges can drastically slow down the training and the inference process of GNNs. Hence, this naïve approach is computationally impractical for GNNs. To address this computational challenge, we will instead propose an efficient homophily booster via a more careful design of "shortcut" edges.

## 3.2 EFFICIENT GRAPH HOMOPHILY BOOSTER

To address the computational inefficiency of the motivating naïve approach above, we propose an efficient, simple yet effective graph transformation method called *GRAph homoPHIly boosTEr* (GRAPHITE) in this subsection.

Note that the large number of NHB "shortcut" edges is because NHB *directly* connects nodes with similar features. Since there are $O(|\mathcal{V}|^2)$ node pairs in a graph, then the total number of added NHB "shortcut" edges can be as large as $O(|\mathcal{V}|^2)$.

To reduce the number of "shortcut" edges, we propose to connect such node pairs *indirectly* instead. In particular, if we can create a few auxiliary "hub" nodes so that all such node pairs are *indirectly* connected through the "hub" nodes, then we will be able to significantly reduce the number of "shortcut" edges at only a small price of adding a few "hub" nodes. Therefore, we need to develop an appropriate design of the "hub" nodes.

**Graph transformation.** Following the aforementioned motivation, we propose to create a *feature node* $x_k$ for each feature $k$ to serve as the "hub" nodes. Let $\mathcal{V}_\mathcal{X}$ denote the set of feature nodes:

$$\mathcal{V}_\mathcal{X} := \{x_k : k \in \mathcal{X}\}. \tag{3}$$

To distinguish feature nodes $\mathcal{V}_\mathcal{X}$ from nodes $\mathcal{V}$ in the original graph, we call $\mathcal{V}$ *graph nodes* from now on. For each graph node $v_i \in \mathcal{V}$, if graph node $v_i$ has feature $k$ (i.e., $\boldsymbol{X}[v_i, k] = 1$), we add an

edge $(v_i, x_k)$ to connect the graph node $v_i$ and the feature node $x_k \in \mathcal{V}_\mathcal{X}$, and we call it a *feature edge*. Let $\mathcal{E}_\mathcal{X}$ denote the set of feature edges:

$$\mathcal{E}_\mathcal{X} := \{(v_i, x_k) : v_i \in \mathcal{V}, \ x_k \in \mathcal{V}_\mathcal{X}, \ \boldsymbol{X}[v_i, k] = 1\} \subseteq \mathcal{V} \times \mathcal{V}_\mathcal{X}.$$

To distinguish feature edges $\mathcal{E}_\mathcal{X}$ from the original edges $\mathcal{E}$, we call $\mathcal{E}$ *graph edges* from now on.

Finally, we define the transformed graph $\mathcal{G}^* = (\mathcal{V}^*, \mathcal{E}^*, \boldsymbol{X}^*)$ as follows. The nodes $\mathcal{V}^*$ of the transformed graph $\mathcal{G}^*$ are the original graph nodes $\mathcal{V}$ and the added feature nodes $\mathcal{V}_\mathcal{X}$: $\mathcal{V}^* := \mathcal{V} \cup \mathcal{V}_\mathcal{X}$. The edges $\mathcal{E}^*$ of the transformed graph $\mathcal{G}^*$ are the original graph edges $\mathcal{E}$ and the added feature edges $\mathcal{E}_\mathcal{X}$: $\mathcal{E}^* := \mathcal{E} \cup \mathcal{E}_\mathcal{X}$. We can also equivalently define the edges of the transformed graph $\mathcal{G}^*$ by its adjacency matrix. Let $\boldsymbol{A}$ denote the adjacency matrix of the original graph $\mathcal{G}$. Then, the adjacency matrix $\boldsymbol{A}^*$ of the transformed graph $\mathcal{G}^*$ can be expressed in a block matrix form:

$$\boldsymbol{A}^* = \begin{bmatrix} \boldsymbol{A} & \boldsymbol{X} \\ \boldsymbol{X}^\top & \boldsymbol{0} \end{bmatrix}. \tag{4}$$

It remains to define node features $\boldsymbol{X}^* \in \mathbb{R}^{\mathcal{V}^* \times \mathcal{X}}$ of the transformed graph. For each graph node $v_i \in \mathcal{V}$, we use its original features as its node features: $\boldsymbol{X}^*[v_i, :] := \boldsymbol{X}[v_i, :]$. For each feature node $x_k \in \mathcal{V}_\mathcal{X}$, we define its node feature as the average feature vector among the graph nodes $v_i$ that are connected to feature node $x_k$:

$$\boldsymbol{X}^*[x_k, :] := \frac{1}{|\mathcal{E}_\mathcal{X} \cap (\mathcal{V} \times \{x_k\})|} \sum_{v_i : (v_i, x_k) \in \mathcal{E}_\mathcal{X}} \boldsymbol{X}[v_i, :]. \tag{5}$$

Our proposed graph transformation GRAPHITE is illustrated in Figure 1. In this example, $\{v_1, v_2, v_3, v_4, v_5\}$ are the graph nodes, where $v_1, v_2$ belong to one class, and $v_3, v_4, v_5$ belong to the other class. Our proposed GRAPHITE adds feature nodes $x_1, x_2, x_3$ to the graph. We can see that feature node $x_1$ facilitates homophilic message passing between $v_1, v_2$, and that feature node $x_2$ facilitates homophilic message passing among $v_3, v_4, v_5$.

**Theoretical guarantees.** The transformed graph $\mathcal{G}^*$ enjoys a few theoretical guarantees. First, an important property of the feature edges is that every pair of nodes that share features can be connected through feature edges within two hops, as formally stated in Observation 2. This ensures that nodes with similar features are close to each other on the transformed graph $\mathcal{G}^*$, facilitating homophilic message passing.

**Observation 2** (two-hop indirect connection). *For each pair of nodes $u, v \in \mathcal{V}$, if they share at least a feature (i.e., $\|\boldsymbol{X}[v_i, :] \wedge \boldsymbol{X}[v_j, :]\|_\infty > 0$), then $v_i$ and $v_j$ are two-hop neighbors on the transformed graph $\mathcal{G}^*$.*

Furthermore, we theoretically show that our proposed graph transformation GRAPHITE can increase the homophily of the graph without increasing the size of the graph much, as formally stated in Theorem 3.

**Theorem 3** (efficient homophily booster). *Given a heterophilic graph $\mathcal{G} = (\mathcal{V}, \mathcal{E}, \boldsymbol{X})$, let $\mathcal{G}^* := (\mathcal{V}^*, \mathcal{E}^*, \boldsymbol{X}^*)$ denote the graph transformed by our proposed GRAPHITE. Under mild and realistic assumptions in Appendix D.1, we have*

$$\mathrm{hom}(\mathcal{G}^*) > \mathrm{hom}(\mathcal{G}), \tag{6}$$

$$|\mathcal{V}^*| \leq O(|\mathcal{V}|), \quad |\mathcal{E}^*| \leq O(|\mathcal{E}|). \tag{7}$$

The effectiveness of our proposed GRAPHITE is also empirically validated in Section 4.3. As shown in Figure 2, our proposed GRAPHITE significantly increases the homophily of originally heterophilic graph.

### 3.3 NEURAL ARCHITECTURE

The transformed graph $\mathcal{G}^*$ can be readily fed into existing GNNs to boost their performance, even when the GNNs were originally designed for homophilic graphs, as demonstrated in Table 4. Meanwhile, to maximize the GNN performance on the transformed graph $\mathcal{G}^*$, we introduce a GNN architecture specially designed for the transformed graph in this subsection.

Table 1: Summary of dataset statistics. We use four heterophilic graphs and two homophilic graphs.

| Statistic | Heterophilic Graphs | | | | Homophilic Graphs | |
|---|---|---|---|---|---|---|
| | ACTOR | SQUIRREL-F | CHAMELEON-F | MINESWEEPER | CORA | CITESEER |
| # Nodes | 7600 | 2223 | 890 | 10000 | 2708 | 3327 |
| # Edges | 33544 | 46998 | 8854 | 39402 | 5429 | 4732 |
| # Features | 931 | 2089 | 2325 | 7 | 1433 | 3703 |
| # Classes | 5 | 5 | 5 | 2 | 7 | 6 |
| Homophily | 0.0028 | 0.0086 | 0.0295 | 0.0094 | 0.7711 | 0.6707 |

To help the GNN distinguish graph nodes $\mathcal{V}$ from feature nodes $\mathcal{V}_{\mathcal{X}}$, we use different edge weights for different edges. As a reference weight, suppose that graph edges $\mathcal{E}$ have weight $w_{\mathcal{E}} := 1$. Let $w_{\mathcal{X}} > 0$ denote the weight of feature edges $\mathcal{E}_{\mathcal{X}}$. Following GCN (Kipf & Welling, 2016), we also use self-loops in GNN message passing; let $w_0 > 0$ denote the weight of self-loops.

Let $d_u$ denote the weighted degree of each node $u \in \mathcal{V}^*$. Specifically, for each graph node $v_i \in \mathcal{V}$,

$$d_{v_i} := w_0 + \sum_{(v_i, v_j) \in \mathcal{E}} w_{\mathcal{E}} + \sum_{(v_i, x_k) \in \mathcal{E}_{\mathcal{X}}} w_{\mathcal{X}}; \tag{8}$$

and for each feature node $x_k \in \mathcal{V}_{\mathcal{X}}$,

$$d_{x_k} := w_0 + \sum_{(v_i, x_k) \in \mathcal{E}_{\mathcal{X}}} w_{\mathcal{X}}. \tag{9}$$

Inspired by FAGCN (Bo et al., 2021), we use a self-gating mechanism in GNN aggregation. For each node $u \in \mathcal{V}^*$, let $\boldsymbol{h}_u \in \mathbb{R}^m$ denote the embedding of node $u$ before GNN aggregation, where $m$ is the embedding dimensionality. Then, the self-gating score $\alpha_{u,u'}$ between two nodes $u, u' \in \mathcal{V}^*$ is defined as

$$\alpha_{u,u'} := \tanh\left(\frac{\boldsymbol{a}^{\mathsf{T}}(\boldsymbol{h}_u \parallel \boldsymbol{h}_{u'}) + b}{\tau}\right). \tag{10}$$

where $\parallel$ denotes the concatenation operation, $\boldsymbol{a} \in \mathbb{R}^{2m}$ and $b \in \mathbb{R}$ are learnable parameters, and $\tau > 0$ is a temperature hyperparameter.

Next, we describe our aggregation mechanism. For each node $u \in \mathcal{V}^*$, let $\boldsymbol{h}'_u \in \mathbb{R}^m$ denote the embedding of node $u$ after GNN aggregation. For each graph node $v_i \in \mathcal{V}$, we define

$$\boldsymbol{h}'_{v_i} := \frac{w_0 \alpha_{v_i, v_i}}{\sqrt{d_{v_i}}\sqrt{d_{v_i}}} \boldsymbol{h}_{v_i} + \sum_{(v_i, v_j) \in \mathcal{E}} \frac{\alpha_{v_i, v_j}}{\sqrt{d_{v_i}}\sqrt{d_{v_j}}} \boldsymbol{h}_{v_j} + \sum_{(v_i, x_j) \in \mathcal{E}_{\mathcal{X}}} \frac{w_{\mathcal{X}} \alpha_{v_i, x_k}}{\sqrt{d_{v_i}}\sqrt{d_{x_k}}} \boldsymbol{h}_{x_k}; \tag{11}$$

and for each feature node $x_k \in \mathcal{V}_{\mathcal{X}}$, we define

$$\boldsymbol{h}'_{x_k} := \frac{w_0 \alpha_{x_k, x_k}}{\sqrt{d_{x_k}}\sqrt{d_{x_k}}} \boldsymbol{h}_{x_k} + \sum_{(v_i, x_k) \in \mathcal{E}_{\mathcal{X}}} \frac{w_{\mathcal{X}} \alpha_{v_i, x_k}}{\sqrt{d_{v_i}}\sqrt{d_{x_k}}} \boldsymbol{h}_{v_i}. \tag{12}$$

Furthermore, we add a multi-layer perceptron (MLP) with residual connections after each GNN aggregation. We use the GELU activation function (Hendrycks & Gimpel, 2016).

## 4 EXPERIMENTS

We conduct extensive experiments on both heterophilic and homophilic datasets to answer the following research questions:

**RQ1:** How does our proposed GRAPHITE compare with state-of-the-art methods?

**RQ2:** How much improvement can our GRAPHITE achieve in terms of the graph homophily?

**RQ3:** Can our graph transformation alone enhance the accuracy of homophilic GNNs?

**RQ4:** How should we design the features of feature nodes in our GRAPHITE?

**RQ5:** How efficient is our GRAPHITE?

**RQ6:** How effective is our GRAPHITE under various hyperparameters?

Table 2: Comparison with existing methods. GRAPHITE *significantly* outperforms state-of-the-art methods on heterophilic graphs while achieving *comparable* accuracy with state-of-the-art methods on homophilic graphs. Best results are marked in **bold**, and second best results are underlined.

| Method | Heterophilic Graphs | | | | Homophilic Graphs | |
|---|---|---|---|---|---|---|
| | ACTOR | SQUIRREL-F | CHAMELEON-F | MINESWEEPER | CORA | CITESEER |
| MLP | $35.04 \pm 1.53$ | $33.91 \pm 1.55$ | $38.44 \pm 5.14$ | $50.99 \pm 1.47$ | $75.45 \pm 1.88$ | $71.53 \pm 0.70$ |
| ChebNet | $34.40 \pm 1.18$ | $31.75 \pm 3.42$ | $34.30 \pm 4.33$ | $\underline{91.60} \pm 0.44$ | $81.58 \pm 5.09$ | $65.18 \pm 8.37$ |
| GCN | $30.21 \pm 0.86$ | $35.57 \pm 1.86$ | $40.06 \pm 4.38$ | $72.32 \pm 0.93$ | $87.50 \pm 1.68$ | $75.77 \pm 0.96$ |
| SGC | $29.26 \pm 1.41$ | $38.27 \pm 2.16$ | $41.40 \pm 4.91$ | $72.11 \pm 0.95$ | $88.05 \pm 2.08$ | $75.80 \pm 1.75$ |
| GAT | $28.86 \pm 0.99$ | $32.74 \pm 3.02$ | $40.11 \pm 2.80$ | $87.59 \pm 1.35$ | $87.11 \pm 1.48$ | $76.43 \pm 1.31$ |
| GraphSAGE | $34.95 \pm 1.06$ | $34.43 \pm 2.68$ | $39.33 \pm 4.53$ | $90.54 \pm 0.66$ | $87.90 \pm 1.73$ | $76.43 \pm 1.19$ |
| GIN | $28.29 \pm 1.45$ | $39.51 \pm 2.83$ | $40.17 \pm 4.76$ | $75.89 \pm 2.09$ | $85.65 \pm 2.26$ | $72.55 \pm 1.78$ |
| APPNP | $33.68 \pm 1.26$ | $33.75 \pm 2.31$ | $37.93 \pm 4.33$ | $67.36 \pm 1.08$ | $87.59 \pm 1.68$ | $75.90 \pm 0.91$ |
| GCNII | $34.78 \pm 1.50$ | $35.93 \pm 2.87$ | $41.56 \pm 2.74$ | $88.42 \pm 0.85$ | $87.20 \pm 1.56$ | $73.84 \pm 0.91$ |
| GATv2 | $28.87 \pm 1.39$ | $32.49 \pm 2.51$ | $39.72 \pm 6.60$ | $88.85 \pm 1.16$ | $87.66 \pm 1.52$ | $76.59 \pm 1.19$ |
| MixHop | $35.40 \pm 1.34$ | $30.43 \pm 2.33$ | $37.93 \pm 3.87$ | $89.68 \pm 0.57$ | $84.53 \pm 1.53$ | $76.11 \pm 0.83$ |
| TAGCN | $34.92 \pm 1.19$ | $33.33 \pm 2.37$ | $41.01 \pm 3.77$ | $91.54 \pm 0.56$ | $88.38 \pm 1.95$ | $76.49 \pm 1.41$ |
| DAGNN | $33.15 \pm 1.14$ | $34.72 \pm 2.55$ | $38.94 \pm 3.53$ | $67.87 \pm 1.26$ | $88.27 \pm 1.53$ | $75.81 \pm 0.90$ |
| JKNet | $28.63 \pm 0.94$ | $\underline{40.81} \pm 2.60$ | $40.39 \pm 4.85$ | $81.00 \pm 0.92$ | $86.24 \pm 0.85$ | $73.11 \pm 1.82$ |
| Virtual Node | $30.71 \pm 0.82$ | $38.00 \pm 2.28$ | $41.45 \pm 5.46$ | $72.36 \pm 0.98$ | $87.24 \pm 2.00$ | $69.80 \pm 6.89$ |
| H2GCN | $34.20 \pm 1.47$ | $34.02 \pm 3.15$ | $40.89 \pm 3.13$ | $87.08 \pm 0.82$ | $76.89 \pm 2.25$ | $75.87 \pm 1.02$ |
| FSGNN | $35.60 \pm 1.34$ | $37.28 \pm 2.63$ | $\underline{43.30} \pm 3.62$ | $50.00 \pm 0.00$ | $87.81 \pm 1.96$ | $76.77 \pm 1.13$ |
| ACM-GNN | $34.04 \pm 1.25$ | $33.91 \pm 2.28$ | $39.78 \pm 4.58$ | $86.35 \pm 0.99$ | $88.58 \pm 1.90$ | $76.47 \pm 0.99$ |
| FAGCN | $\underline{36.18} \pm 1.52$ | $36.52 \pm 1.72$ | $39.83 \pm 3.93$ | $84.69 \pm 2.05$ | $88.66 \pm 2.11$ | $\underline{76.82} \pm 1.48$ |
| OrderedGNN | $35.64 \pm 0.98$ | $32.70 \pm 2.42$ | $38.38 \pm 3.65$ | $91.01 \pm 0.50$ | $84.81 \pm 1.67$ | $74.10 \pm 1.62$ |
| GloGNN | $19.80 \pm 2.61$ | $28.72 \pm 2.63$ | $40.17 \pm 4.66$ | $53.42 \pm 1.47$ | $73.02 \pm 2.98$ | $72.46 \pm 2.09$ |
| GGCN | $32.76 \pm 1.39$ | $35.06 \pm 5.65$ | $34.08 \pm 3.44$ | $84.76 \pm 1.84$ | $86.39 \pm 1.93$ | $75.36 \pm 1.99$ |
| GPRGNN | $35.42 \pm 1.33$ | $34.97 \pm 2.83$ | $40.50 \pm 4.55$ | $83.94 \pm 0.98$ | $\mathbf{88.86} \pm 1.42$ | $76.49 \pm 1.00$ |
| ALT | $33.10 \pm 1.38$ | $37.28 \pm 1.49$ | $39.61 \pm 3.36$ | $89.06 \pm 0.64$ | $\underline{88.82} \pm 2.02$ | $\mathbf{76.88} \pm 1.20$ |
| NodeFormer | $29.26 \pm 2.31$ | $24.29 \pm 2.60$ | $34.92 \pm 4.08$ | $77.71 \pm 3.50$ | $87.44 \pm 1.37$ | $75.20 \pm 1.27$ |
| SGFormer | $25.89 \pm 0.80$ | $34.54 \pm 2.96$ | $42.79 \pm 4.06$ | $52.06 \pm 0.50$ | $86.24 \pm 1.58$ | $70.74 \pm 1.25$ |
| DIFFormer | $26.31 \pm 1.19$ | $33.17 \pm 2.84$ | $39.16 \pm 4.10$ | $69.25 \pm 0.93$ | $86.61 \pm 3.04$ | $76.65 \pm 1.52$ |
| **GRAPHITE** (Ours) | $\mathbf{37.69} \pm 1.57$ | $\mathbf{43.06} \pm 2.89$ | $\mathbf{45.08} \pm 4.04$ | $\mathbf{94.78} \pm 0.41$ | $88.23 \pm 1.65$ | $76.41 \pm 1.57$ |

## 4.1 EXPERIMENTAL SETTINGS

**Datasets.** We evaluate GRAPHITE and various baseline methods across six real-world datasets. The dataset statistics are summarized in Table 1. The reported homophily is the *adjusted homophily* introduced in Platonov et al. (2024), which exhibits more desirable properties compared to traditional edge/node homophily. We leverage *adjusted homophily* to categorize the datasets into two groups: *heterophilic* and *homophilic*. Please see Appendix A.1 for dataset descriptions.

**Training and evaluation.** To benchmark GRAPHITE and compare it with the baseline methods, we use *node classification* tasks with performance measured by classification accuracy on ACTOR, CHAMELEON-F, SQUIRREL-F, CORA, and CITESEER and by ROC-AUC on MINESWEEPER following Platonov et al. (2023). For all baseline methods, we use the hyperparameters provided by the authors. For the evaluation on ACTOR, CHAMELEON-F, and SQUIRREL-F, we generate 10 random splits with a ratio of $48\%/32\%/20\%$ as the training/validation/test set, respectively, following Gu et al. (2024). For the evaluation on MINESWEEPER, we directly utilize the 10 random splits provided by the original paper (Platonov et al., 2023). For the evaluation on CORA and CITESEER, we follow Luan et al. (2021) and Chien et al. (2020) to randomly generate 10 random splits with a ratio of $60\%/20\%/20\%$ as the training/validation/test set, respectively. For each experiment, we report the mean and the standard deviation of the performance metric across the corresponding 10 random splits. Please see Appendix A for additional experimental settings.

## 4.2 MAIN RESULTS

To answer RQ1, we compare the proposed method GRAPHITE with 27 state-of-the-art methods on six heterophilic and homophilic graphs. The results are shown in Table 2.

As shown in Table 2, our GRAPHITE achieves significant performance gains (p-value<0.1) over prior state-of-the-art GNN methods on heterophilic graphs while maintaining competitive accuracy on homophilic graphs. Specifically, GRAPHITE outperforms the best baseline methods by $4.17\%, 5.23\%, 5.35\%$ and $3.47\%$ on ACTOR, SQUIRREL-F, CHEMELEON-F and MINESWEEPER, respectively. While some existing models perform well on individual datasets, they often struggle on others, highlighting their insufficient consistency. In contrast, GRAPHITE demonstrates the best results across all four heterophilic benchmarks. Another interesting observation is that while GRAPHITE is built upon FAGCN (Bo et al., 2021), it significantly surpasses FAGCN, demonstrating the beneficial effect of our graph transformation and feature edges.

**Discussion.** It is worth noting that most of the baseline methods cannot achieve better results compared to MLP on ACTOR, which can be explained by the fact that these methods typically treat node features and graph structure as joint input without explicitly decoupling them. The weak structural homophily exhibited by ACTOR makes typical GNNs fail to capture important feature signals, reinforcing the importance of our graph transformation strategy that boosts *feature homophily* significantly. For SQUIRREL-F, we find that JKNet is the best among baselines. This observation reveals that structure information is very important within SQUIRREL-F since JKNet aggregates feature knowledge from multi-hop neighbors to learn structure-aware representation. This finding also explains the success of GRAPHITE since the useful multi-hop information in SQUIRREL-F can be propagated even more efficiently through the constructed *feature edges*.

As another example, SGFormer performs the best on CHAMELEON-F among baseline methods. We argue that CHAMELEON-F needs a considerable amount of global messages and graph transformers are experts at capturing this type of information. Compared with NodeFormer and DIF-Former, SGFormer is the most advanced graph transformer utilizing simplified graph attention that strikes a good balance between global structural information and feature signal, preventing the over-globalizing issue (Xing et al., 2024). Similarly, GRAPHITE transforms the original graph into a form that facilitates global message exchange by the introduction of *feature edges*. As a final remark, although GRAPHITE is designed specifically to deal with heterophilic datasets, GRAPHITE still maintains competitive accuracy on homophilic datasets (CORA and CITESEER), achieving results that are on par with the best existing methods.

## 4.3 HOMOPHILY ANALYSIS

To answer RQ2, we conduct a homophily analysis across heterophilic datasets under two homophily metrics: *feature homophily* $H^{\text{feature}}(\mathcal{G})$ and *adjusted homophily* $H^{\text{adjusted}}(\mathcal{G})$ (see Appendix B for their formal definitions). Table 3 and Figure 2 show the relative improvements between the homophily metrics before and after applying GRAPHITE. We can observe a significant boost in both homophilily metrics after applying GRAPHITE across the four heterophilic datasets.

Table 3: Relative improvement ratio of *feature homophily* and *adjusted homophily* across datasets. GRAPHITE significantly boosts both homophily metrics. See Figure 2 for visualization.

| **Dataset** | $H^{\text{feature}}(\mathcal{G})$ | $H^{\text{adjusted}}(\mathcal{G})$ |
| --- | --- | --- |
| ACTOR | +179% | +2767% |
| SQUIRREL-F | +961% | +215% |
| CHAMELEON-F | +1739% | +402% |
| MINESWEEPER | +41% | +1023% |

**Discussion.** Overall, GRAPHITE effectively boosts both homophily metrics across all heterophilic datasets. Specifically, Squirrel-F and Chameleon-F demonstrate significant boosts in terms of *feature homophily*. This is mainly because their discrete features directly correspond to specific topics and each feature edge will contribute much higher feature similarity than usual edges. On the other hand, Actor and Minesweeper showcase much higher *adjusted homophily* after applying GRAPHITE. For Actor, this favorable behavior can be attributed to the high correlation between page co-occurrences and node labels; while for Minesweeper, the sum of label-specific node degrees (defined in Equation (14)) increases much due to the transformation performed by GRAPHITE.

**Baseline methods.** In our experiments, we consider a wide range of GNN baselines, including MLP (structure-agnostic), homophilic GNNs, heterophilic GNNs, and Graph Transformers. The full list is shown in Appendix A.2.

Table 4: Effectiveness of the proposed graph transformation. GRAPHITE transformed graphs alone can already enhance the performance of homophilic GNNs.

| Dataset
+GRAPHITE? | ACTOR | | MINESWEEPER | |
|---|---|---|---|---|
| | ✗ | ✓ | ✗ | ✓ |
| GCN | $30.21 \pm 0.86$ | $\textbf{34.83} \pm 1.28$ | $72.32 \pm 0.93$ | $\textbf{75.38} \pm 1.56$ |
| GAT | $28.86 \pm 0.99$ | $\textbf{32.09} \pm 1.35$ | $87.59 \pm 1.35$ | $\textbf{88.66} \pm 0.88$ |
| GraphSAGE | $34.95 \pm 1.06$ | $\textbf{35.09} \pm 1.06$ | $90.54 \pm 0.66$ | $\textbf{90.85} \pm 0.67$ |
| JKNet | $28.63 \pm 0.94$ | $\textbf{35.96} \pm 1.40$ | $81.00 \pm 0.92$ | $\textbf{85.56} \pm 2.59$ |
| GIN | $28.29 \pm 1.45$ | $\textbf{33.75} \pm 1.83$ | $75.89 \pm 2.09$ | $\textbf{87.07} \pm 1.71$ |

Table 5: Comparison of aggregators for the features of feature nodes. All aggregators in fact perform similarly, so we choose averaging due to its simplicity and efficiency.

| Dataset | Averaging (Ours) | Learned Embeddings | Learned Attention | Majority Voting |
|---|---|---|---|---|
| ACTOR | 37.69 | 37.46 | 37.13 | 37.59 |
| SQUIRREL-F | 43.06 | 43.53 | 43.46 | 43.28 |
| CHAMELEON-F | 45.08 | 44.80 | 44.36 | 45.64 |
| MINESWEEPER | 94.78 | 94.47 | 94.56 | 94.75 |

## 4.4 ABLATION STUDIES

**Graph transformation.** To further demonstrate the effectiveness of our proposed graph transformation GRAPHITE and answer RQ3, we compare the performance of homophilic GNNs on the original graph and that on the transformed graph. In this experiment, we use two larger-scale datasets, ACTOR and MINESWEEPER, and five representative homophilic GNNs, GCN, GAT, GraphSAGE, JKNet, and GIN. The results are presented in Table 4.

From Table 4, we can see that our proposed GRAPHITE consistently improves the performance of the five representative homophilic GNNs on both datasets, even though these GNNs are not specially designed for modeling feature nodes. For example, the accuracy of GAT on ACTOR is enhanced from 30.21% to 34.83%, which is a relative improvement of 15.29%. The results demonstrate that our proposed graph transformation GRAPHITE can significantly enhance the performance of homophilic GNNs on originally heterophilic graphs, echoing the fact that our proposed graph transformation can significantly increase the graph homophily.

**Features of feature nodes.** GRAPHITE uses averaging aggregation to define the features of feature nodes. To elaborate on the rationale of this simple aggregator and answer RQ4, we compare with other aggregators on heterophilic datasets. The results are shown in Table 5. In fact, all aggregators have similar accuracies while averaging is more efficient than learned embeddings and attention-weighted aggregation and is simpler than majority voting. Therefore, we use averaging in our method due to its simplicity and efficiency.

**Computational efficiency.** To evaluate the efficiency of our method and answer RQ5, we provide a comparison of running times with and without our transformation, respectively. The results are shown in Table 6 (sorted by graph sizes). We can see that our graph transformation has only a small impact on running time while significantly improving accuracy. Notably, the computational overhead gets smaller on larger graphs (e.g., Minesweeper), justifying the scalability of our method. This is because the number of added nodes (i.e., the number of features) is typically negligibly small compared with the number of nodes on large graphs, and the number of feature edges is proportional to the space complexity of the node feature matrix $\boldsymbol{X}$.

**Hyperparameters.** Since our method has a few hyperparameters including $\tau$ and $w_{\mathcal{X}}$, to answer RQ6, we provide a sensitivity analysis of them on MINESWEEPER. The results are shown in Table 7. From the table, we can see that our method is not sensitive to these hyperparameters, and our method consistently outperform the best baseline under various hyperparameter values.

## 5 RELATED WORK

A substantial body of research has explored the challenges of heterophily in graph neural networks (GNNs). Many early approaches sought to improve information aggregation, such as MixHop (Abu-

Table 6: Running time comparison with and without GRAPHITE. Graphs are sorted in decreasing order by graph sizes. Our graph transformation has only a small impact on running time while significantly improves accuracy.

| Method | MINESWEEPER | ACTOR | SQUIRREL-F | CHAMELEON-F |
|---|---|---|---|---|
| No transformation | 1.9 min | 1.5 min | 0.7 min | 0.5 min |
| With transformation | 2.3 min | 2.0 min | 1.1 min | 0.7 min |

Table 7: Sensitivity analysis of hyperparameters $\tau$ and $w_{\mathcal{X}}$ on MINESWEEPER. Our method consistently outperform the best baseline under various hyperparameters.

| Best Baseline | GRAPHITE (Ours) | | | | |
|---|---|---|---|---|---|
| | $\tau = 0.1$ | $\tau = 0.5$ | $\tau = 1.0$ | $\tau = 1.5$ | $\tau = 2.0$ |
| 91.60 | 93.48 | 94.29 | 94.78 | 94.66 | 94.18 |
| Best Baseline | GRAPHITE (Ours) | | | | |
| | $w_{\mathcal{X}} = 0.1$ | $w_{\mathcal{X}} = 0.25$ | $w_{\mathcal{X}} = 0.5$ | $w_{\mathcal{X}} = 0.75$ | $w_{\mathcal{X}} = 1.0$ |
| 91.60 | 94.78 | 94.16 | 93.95 | 93.53 | 93.15 |

El-Haija et al., 2019), which mixes different-hop neighborhood features, and GPRGNN (Chien et al., 2020), which employs generalized PageRank propagation for adaptive message passing. Other methods focus on explicit heterophilic adaptations, such as H2GCN (Zhu et al., 2020), which separates ego- and neighbor-embeddings, and FAGCN (Bo et al., 2021), which learns optimal representations via frequency-adaptive filtering. Additional works, including OrderedGNN (Song et al., 2023), GloGNN (Li et al., 2022), and GGCN (Yan et al., 2022), leverage structural ordering, global context, and edge corrections, respectively, to enhance performance on heterophilic graphs. Recent advances explore alternative formulations, such as component-wise signal decomposition (e.g. ALT, Xu et al., 2023) and adaptive residual mechanisms (Xu et al., 2024a; Yan et al., 2024) for greater flexibility. Beyond architectural innovations, rigorous benchmarking efforts (Lim et al., 2021; Zhu et al., 2024; Platonov et al., 2023) have been introduced to standardize evaluations and assess generalization across diverse graph properties. A broader synthesis of heterophilic GNN techniques can be found in recent surveys (Zheng et al., 2022; Zhu et al., 2023; Luan et al., 2024; Gong et al., 2024). Please refer to Appendix C for additional related work.

## 6 CONCLUSION & FUTURE WORK

In this paper, we propose GRAPHITE, a simple yet efficient framework to address the heterophily issue in node classification. By introducing feature nodes that connect to graph nodes with corresponding discrete features, we can solve the heterophily issue by increasing the graph homophily ratio. Through theoretical analysis and empirical study, we validate that GRAPHITE can indeed effectively increase the graph homophily. Our extensive experiments demonstrate that GRAPHITE consistently outperforms state-of-the-art methods on heterophilic graph datasets and achieves comparable performance on homophilic graph datasets. An interesting future direction would be extending the proposed graph transformation to general graphs with continuous node features; possible approaches include clustering the continuous features into a few clusters and binning the continuous features into discrete buckets. Other future directions include (i) studying how our graph transformation affects graph properties, (ii) connecting to the node distinguishability principle (Luan et al., 2023), and (iii) identifying an optimal subset of features (Zheng et al., 2025).

### ACKNOWLEDGEMENTS

This work is supported by NSF (2416070). The content of the information in this document does not necessarily reflect the position or the policy of the Government, and no official endorsement should be inferred. The U.S. Government is authorized to reproduce and distribute reprints for Government purposes notwithstanding any copyright notation hereon.

ETHICS STATEMENT

Our study is based entirely on publicly available graph datasets commonly used in the GNN literature and does not involve private or sensitive information. We develop a graph transformation framework that explicitly increases graph homophily to enable more effective message passing. To ensure methodological soundness and reproducibility, we provide both theoretical analyses and extensive empirical evaluations across heterophilic datasets. The release of code and data splits is intended solely for academic research to advance the understanding of graph machine learning and to support future work on graph neural networks, and are not designed for sensitive or high-stakes applications.

REPRODUCIBILITY STATEMENT

We include the conceptual framework, transformation steps, method details and evaluation setup in the paper and appendix. To facilitate reproducibility, we also publicly release our code at `https://github.com/q-rz/ICLR26-GRAPHITE`.

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

# A  EXPERIMENTAL SETTINGS (CONT'D)

## A.1  DATASETS (CONT'D)

For heterophilic group, we consider the following datasets, which are widely used as benchmarks for studying graph learning methods under heterophilic settings.

- ACTOR (Pei et al., 2020): ACTOR dataset is an actor-only induced subgraph of the film dataset introduced by (Tang et al., 2009). The nodes are actors and the edges denote co-occurrence on the same Wikipedia page. The node features are keywords on the pages and we classify nodes into five categories.

- Squirrel-Filtered (SQUIRREL-F, Platonov et al., 2023): SQUIRREL-F is a page-page dataset. It is a subset of the Wiki dataset (Rozemberczki et al., 2021) that focus on the topic related to squirrel. Nodes are web pages and edges are mutual links between pages. The node features are important keywords in the pages and we classify nodes into five categories in terms of traffic of the webpage.

- Chameleon-Filtered (CHAMELEON-F, Platonov et al., 2023): CHAMELEON-F is a page-page dataset. It is a subset of the Wiki dataset (Rozemberczki et al., 2021) that focus on the topic related to chameleon. Nodes are web pages and edges are mutual links between pages. The node features are important keywords in the pages and we classify nodes into five categories in terms of traffic of the webpage.

- MINESWEEPER (Platonov et al., 2023): MINESWEEPER is a synthetic dataset that simulates a Minesweeper game with $100 \times 100$ grid. Each node is connected to its neighboring nodes where $20\%$ nodes are selected as mines at random. Node features are numbers of neighboring mines and the goal is to predict whether each test node is mine. These datasets are widely used as benchmarks for studying graph learning methods under heterophilic settings.

For the homophilic group, we consider the following datasets, which are standard homophilic network benchmarks.

- CORA (Sen et al., 2008) : Cora dataset is a citation network, where nodes represent scientific papers in the machine learning field, and edges correspond to citation relationships between these papers. Each node is associated with a set of features that describe the paper, represented as a bag-of-words model. The task for this dataset is to classify each paper into one of seven categories, reflecting the area of research the paper belongs to.

- CITESEER (Sen et al., 2008): CiteSeer dataset is a citation network of scientific papers. It consists of research papers as nodes, with citation links forming the edges between them. Each node is associated with a set of features derived from the paper's content, which is a bag-of-words representation of the paper's text. The task for this dataset is to classify each paper into one of six categories, each representing a specific field of study.

## A.2  BASELINE METHODS (CONT'D)

We briefly introduce GNN-based baseline methods as follows.

The first category is *homophilic GNNs*, which are originally designed under the homophily assumption.

- ChebNet (Defferrard et al., 2016): Uses Chebyshev polynomials to approximate graph convolutions.

- GCN (Kipf & Welling, 2016): Employs a first-order Chebyshev approximation for spectral graph convolutions.

- SGC (Wu et al., 2019): Simplifies GCN by removing non-linearities and collapsing weight matrices for efficiency.

- GAT (Veličković et al., 2018): Introduces attention mechanisms to assign adaptive importance to edges.

- GraphSAGE (Hamilton et al., 2017): Uses several aggregators for inductive graph learning.
- GIN (Xu et al., 2018a): Employs sum-based aggregation to maximize graph structure expressiveness.
- APPNP (Gasteiger et al., 2019): Combines personalized PageRank with neural propagation.
- GCNII (Chen et al., 2020): Extends GCN with residual connections and identity mapping for deep GNN training.
- GATv2 (Brody et al., 2021): Enhances GAT with dynamic attention coefficients for flexible neighbor weighting.
- MixHop (Abu-El-Haija et al., 2019): Aggregates multi-hop neighborhood features by mixing different powers of adjacency matrices.
- TAGCN (Du et al., 2017): Introduces trainable polynomial filters for adaptive, multi-scale feature extraction.
- DAGNN (Liu et al., 2020): Uses dual attention to decouple message aggregation and transformation, improving depth scalability.
- JKNet (Xu et al., 2018b): Uses a jumping knowledge mechanism to combine features from different layers adaptively. We default the backbone GNN model to GCN.
- Virtual Node (Gilmer et al., 2017): Introduces an auxiliary global node to facilitate message passing. We default the backbone GNN model to GCN.

The second category is *heterophilic GNN*s, which are designed for graphs where connected nodes often have different labels.

- H2GCN (Zhu et al., 2020): Enhances GNNs by ego-/neighbor-embedding seperation, higher-order neighbors and intermediate representation combinations.
- FSGNN (Maurya et al., 2021): Employs soft feature selection and hop normalization over GNN layers to form a simple, shallow GNN. We use their default 3-hop variant.
- ACM-GNN (Luan et al., 2022): Introduces adaptive channel mixing to diversify local information. We use their default ACM-GCN+ variant.
- FAGCN (Bo et al., 2021): Uses frequency adaptive filtering to learn optimal graph representations.
- OrderedGNN (Song et al., 2023): Aligns the order to encode neighborhood information and avoids feature mixing.
- GloGNN (Li et al., 2022): Incorporates global structural information to enhance graph learning beyond local neighborhoods.
- GGCN (Yan et al., 2022): Utilizes structure/feature-based edge correction to combat oversmoothing and heterophily.
- GPRGNN (Chien et al., 2020): Introduces generalized PageRank propagation to capture the graph structure.
- ALT (Xu et al., 2023): Decomposes graph into components, extracts signals from these components, and adaptively integrate these signals.

The last category is *graph transformers*, which adapt transformer architectures to graph data and look beyond local neighborhood aggregation.

- NodeFormer (Wu et al., 2022): Introduces all-pair message passing on layer-specific adaptive latent graphs, enabling global feature propagation with linear complexity.
- SGFormer (Wu et al., 2024a): Develops a graph encoder backbone that efficiently computes all-pair interactions with one-layer attentive propagation.
- DIFFormer (Wu et al., 2023): Proposes an energy-constrained diffusion model, leading to variants that are efficient and capable of capturing complex structures.

## A.3 Training & Evaluation (Cont'd)

For our method, we use $w_{\mathcal{X}} \in \{0.01, 0.1, 0.6, 8\}$, $w_0 \in \{0.1, 0.2, 0.3, 0.5, 1, 8\}$, $\tau \in \{0.01, 0.1, 1\}$, and dropout rate 0.2. We use the GNN architecture described in the method section with 8 GNN layers with hidden dimensionality 512 and add a two-layer MLP after each GNN layer for heterophilic graphs and use FAGCN for homophilic graphs. We use original node features as described in Section 3.2, except that we use zeros as the features of graph nodes on Squirrel-F and that we normalize the features of graph nodes on Cora and CiteSeer after computing the features of feature nodes. We train the GNN with learning rate 0.00003 for 1000 steps using the Adam optimizer (Kingma & Ba, 2014). Our method was implemented in PyTorch 2.7.0 and Deep Graph Library (DGL) 2.4.0, and experiments were run on Intel Xeon CPU @ 2.20GHz with 96GB memory and NVIDIA Tesla V100 32GB GPU.

## B Definition of Homophily Metrics

To measure to what extent GRAPHITE can boost graph homophily on heterophlic datasets, we consider two popular homophily metrics: *feature homophily* (Jin et al., 2022) and *adjusted homophily* (Platonov et al., 2024). Formally, given a graph $\mathcal{G}$, *feature homophily* $H^{\text{feature}}$ is defined as follows:

$$H^{\text{feature}}(\mathcal{G}) := \frac{1}{|\mathcal{E}|} \sum_{(v_i, v_j) \in \mathcal{E}} \text{sim}(v_i, v_j), \tag{13}$$

where $\text{sim}(v_i, v_j) := \cos(\boldsymbol{X}[v_i, :], \boldsymbol{X}[v_j, :])$ is the cosine-similarity computed between features of nodes $v_i, v_j$. This metric is a variant of the *generalized edge homophily ratio* $H^{\text{edge}}$ proposed by (Jin et al., 2022), which measures the feature similarity between each of the connected node pairs in the graph dataset. Then, the *adjusted homophily* $H^{\text{adjusted}}$ is defined as follows:

$$H^{\text{adjusted}}(\mathcal{G}) := \frac{H^{\text{edge}}(\mathcal{G}) - \sum_{c=1}^{C} D_c^2 / (2|\mathcal{E}|)^2}{1 - \sum_{c=1}^{C} D_c^2 / (2|\mathcal{E}|)^2}, \tag{14}$$

where $C$ denotes the number of classes and $H^{\text{edge}}(\mathcal{G})$ is *edge homophily*, which is defined similarly as Equation (13) with the similarity function $\text{sim}(v_i, v_j) = \mathbf{1}_{\{y_{v_i} = y_{v_j}\}}$, and

$$D_c := \sum_{v \in \mathcal{V}} \deg(v) 1_{[y_v = c]} \tag{15}$$

is the sum of degrees $\deg(v)$ of nodes with label $c$, where $y_v$ denotes the label of node $v$. Since we do not have node labels for the *feature nodes* when computing *adjusted homophily*, we assign them a "soft label," which is a uniform probability distribution over the labels of its 1-hop neighbors.

## C Additional Related Work

In modern machine learning research (Ma et al., 2022; Yu et al., 2026; 2025; Zhang et al., 2026; Bao et al., 2025; Chen et al., 2024; Wei et al., 2026a;b; 2025a;b; 2024; 2022; Cui et al., 2026; Chen et al., 2026; Liu et al., 2025a;b; 2024a;b;c; 2023; Bartan et al., 2025; Zeng et al., 2026a;b;c; 2025a;b; 2024a;b; 2023a;b; Zou et al., 2025a;b; Lin et al., 2026; 2025; 2024; Zhou et al., 2025; Jing et al., 2025; Qiu et al., 2026; 2025a;b;c; 2024; 2023; 2022; Xu et al., 2024b; Li et al., 2025a;b;c;d; 2023b; Yoo et al., 2025a;b; 2024; Chan et al., 2024; Wu et al., 2024b; He et al., 2026; 2024; Wang et al., 2023), a problem related to heterophily is over-squashing. The over-squashing problem in message passing neural networks arises when long-range information is exponentially compressed, preventing effective dissemination across the graph (Alon & Yahav, 2020; Shi et al., 2023b). A primary research direction addresses this issue by identifying topological bottlenecks and modifying graph connectivity. Topping et al. (2021) established an initial framework linking oversquashing to graph Ricci curvature, demonstrating that negatively curved edges act as bottlenecks. Building on this idea, subsequent works have developed rewiring strategies inspired by curvature-based principles (Nguyen et al., 2023; Shi et al., 2023a). Beyond curvature, Black et al. (2023) introduced a perspective using effective resistance. Another line of research leverages spectral methods to counteract over-squashing, with notable approaches including spectral gaps (Arnaiz-Rodríguez et al.,

2022), expander graph constructions (Deac et al., 2022), and first-order spectral rewiring (Karhad-kar et al., 2022). More recently, Di Giovanni et al. (2023) provided a comprehensive analysis of the factors contributing to oversquashing. Additional solutions explore advanced rewiring strategies and novel message-passing paradigms (Barbero et al., 2023; Qian et al., 2023; Behrouz & Hashemi, 2024).

# D   THEORETICAL ANALYSIS

## D.1   ASSUMPTIONS

In this subsection, we introduce the assumptions of our theoretical analysis, which are mild and realistic.

Given a graph $\mathcal{G} = (\mathcal{V}, \mathcal{E}, \boldsymbol{X})$ with $\mathcal{E} \neq \varnothing$ and $\boldsymbol{X} \in \{0,1\}^{\mathcal{V} \times \mathcal{X}}$, we define the feature similarity metric as $\mathrm{sim}(v_i, v_j) := \|\boldsymbol{X}[v_i,:] \wedge \boldsymbol{X}[v_j,:]\|_\infty$ and use the feature homophily as the homophily metric:

$$\mathrm{hom}(\mathcal{G}) := \frac{1}{|\mathcal{E}|} \sum_{(v_i, v_j) \in \mathcal{E}} \mathrm{sim}(v_i, v_j). \tag{16}$$

Furthermore, we assume that the original graph $\mathcal{G}$ is heterophilic. That is, we have $\mathrm{hom}(\mathcal{G}) < 1$ while there exists a pair of nodes, $v_i, v_j \in \mathcal{V}$ ($v_i \neq v_j$), such that $\mathrm{sim}(v_i, v_j) > 0$ but $(v_i, v_j) \notin \mathcal{E}$.

Besides that, we assume that the given graph $\mathcal{G}$ does not have too dense features. Formally, we assume that $|\mathcal{X}| \leq O(|\mathcal{V}|)$ and that $\|\boldsymbol{X}\|_0 \leq O(|\mathcal{E}|)$. For the transformed graph $\mathcal{G}^*$, we assume that every feature is used: for any feature $k \in \mathcal{X}$, there exists a graph node $v_i \in \mathcal{V}$ such that $\boldsymbol{X}[v_i, k] = 1$.

## D.2   TECHNICAL LEMMA

Here, we prove a technical lemma that we will use later.

**Lemma 4.** *Let $\mathcal{A}, \mathcal{B} \subset \mathbb{R}$ be two nonempty, finite multisets such that*

$$\frac{1}{|\mathcal{A}|} \sum_{z \in \mathcal{A}} z < \frac{1}{|\mathcal{B}|} \sum_{z \in \mathcal{B}} z.$$

*Then,*

$$\frac{1}{|\mathcal{A} \sqcup \mathcal{B}|} \sum_{z \in \mathcal{A} \sqcup \mathcal{B}} z > \frac{1}{|\mathcal{A}|} \sum_{z \in \mathcal{A}} z.$$

*Proof.* To simplify notation, let

$$\mu_A := \frac{1}{|\mathcal{A}|} \sum_{z \in \mathcal{A}} z, \tag{17}$$

$$\mu_B := \frac{1}{|\mathcal{B}|} \sum_{z \in \mathcal{B}} z, \tag{18}$$

$$\Delta := \mu_{\mathcal{B}} - \mu_{\mathcal{A}} > 0. \tag{19}$$

Then,

$$\frac{1}{|\mathcal{A} \sqcup \mathcal{B}|} \sum_{z \in \mathcal{A} \sqcup \mathcal{B}} z - \frac{1}{|\mathcal{A}|} \sum_{z \in \mathcal{A}} z \tag{20}$$

$$= \frac{1}{|\mathcal{A}| + |\mathcal{B}|} \left( \sum_{z \in \mathcal{A}} z + \sum_{z \in \mathcal{B}} z \right) - \frac{1}{|\mathcal{A}|} \sum_{z \in \mathcal{A}} z \tag{21}$$

$$= \frac{1}{|\mathcal{A}| + |\mathcal{B}|} \left( |\mathcal{A}| \cdot \frac{1}{|\mathcal{A}|} \sum_{z \in \mathcal{A}} z + \sum_{z \in \mathcal{B}} z \right) - \frac{1}{|\mathcal{A}|} \sum_{z \in \mathcal{A}} z \tag{22}$$

$$= \frac{1}{|\mathcal{A}| + |\mathcal{B}|} \left( |\mathcal{A}| \cdot \mu_{\mathcal{A}} + \sum_{z \in \mathcal{B}} z \right) - \mu_{\mathcal{A}} \tag{23}$$

$$= \frac{1}{|\mathcal{A}| + |\mathcal{B}|} \left( |\mathcal{A}| \cdot \mu_{\mathcal{A}} + |\mathcal{B}| \cdot \mu_{\mathcal{B}} \right) - \mu_{\mathcal{A}} \tag{24}$$

$$= \frac{1}{|\mathcal{A}| + |\mathcal{B}|} \left( |\mathcal{A}| \cdot \mu_{\mathcal{A}} + |\mathcal{B}| \cdot \mu_{\mathcal{B}} - (|\mathcal{A}| + |\mathcal{B}|) \cdot \mu_{\mathcal{A}} \right) \tag{25}$$

$$= \frac{1}{|\mathcal{A}| + |\mathcal{B}|} \left( |\mathcal{B}| \cdot \mu_{\mathcal{B}} - |\mathcal{B}| \cdot \mu_{\mathcal{A}} \right) \tag{26}$$

$$= \frac{|\mathcal{B}|}{|\mathcal{A}| + |\mathcal{B}|} \left( \mu_{\mathcal{B}} - \mu_{\mathcal{A}} \right) \tag{27}$$

$$= \frac{|\mathcal{B}|}{|\mathcal{A}| + |\mathcal{B}|} \Delta > 0. \tag{28}$$

It follows that

$$\frac{1}{|\mathcal{A} \sqcup \mathcal{B}|} \sum_{z \in \mathcal{A} \sqcup \mathcal{B}} z > \frac{1}{|\mathcal{A}|} \sum_{z \in \mathcal{A}} z. \qquad \square$$

### D.3 PROOF OF THEOREM 1

**Homophily.** Since the original graph $\mathcal{G}$ is homophilic, then there exists a pair of nodes, $v_i, v_j \in \mathcal{V}$ ($v_i \neq v_j$), such that $\text{sim}(v_i, v_j) = \|\boldsymbol{X}[v_i, :] \wedge \boldsymbol{X}[v_j, :]\|_\infty > 0$ but $(v_i, v_j) \notin \mathcal{E}$. According to the definition of $\mathcal{E}^\dagger$, we know that $(v_i, v_j) \in \mathcal{E}^\dagger \setminus \mathcal{E} \neq \varnothing$, so $\mathcal{E}^\dagger \setminus \mathcal{E} \neq \varnothing$.

Furthermore, for any $(v_i, v_j) \in \mathcal{E}^\dagger \setminus \mathcal{E}$, since $\text{sim}(v_i, v_j) = \|\boldsymbol{X}[v_i, :] \wedge \boldsymbol{X}[v_j, :]\|_\infty > 0$, then there exists a feature $k \in \mathcal{X}$ such that $\boldsymbol{X}[v_i, k] \wedge \boldsymbol{X}[v_j, k] > 0$. Since the feature matrix $\boldsymbol{X}$ is binary, then we must have

$$\boldsymbol{X}[v_i, k] = 1, \qquad \boldsymbol{X}[v_j, k] = 1. \tag{29}$$

It follows that

$$\text{sim}(v_i, v_j) = \|\boldsymbol{X}[v_i, :] \wedge \boldsymbol{X}[v_j, :]\|_\infty \tag{30}$$

$$= \max_{k' \in \mathcal{X}} |\boldsymbol{X}[v_i, k'] \wedge \boldsymbol{X}[v_j, k']| \tag{31}$$

$$\geq |\boldsymbol{X}[v_i, k] \wedge \boldsymbol{X}[v_j, k]| \tag{32}$$

$$= |1 \wedge 1| = 1. \tag{33}$$

Since $\text{hom}(\mathcal{G}) < 1$, then

$$\text{sim}(v_i, v_j) \geq 1 > \text{hom}(\mathcal{G}). \tag{34}$$

Therefore, by Lemma 4 with

$$\mathcal{A} := \{\!\{\text{sim}(v_i, v_j) : (v_i, v_j) \in \mathcal{E}\}\!\}, \tag{35}$$

$$\mathcal{B} := \{\!\{\text{sim}(v_i, v_j) : (v_i, v_j) \in \mathcal{E}^\dagger \setminus \mathcal{E}\}\!\}, \tag{36}$$

we have

$$\text{hom}(\mathcal{G}^\dagger) = \frac{1}{|\mathcal{E}^\dagger|} \sum_{(v_i, v_j) \in \mathcal{E}^\dagger} \text{sim}(v_i, v_j) \tag{37}$$

$$= \frac{1}{|\mathcal{E} \sqcup (\mathcal{E}^\dagger \setminus \mathcal{E})|} \sum_{(v_i, v_j) \in \mathcal{E} \sqcup (\mathcal{E}^\dagger \setminus \mathcal{E})} \text{sim}(v_i, v_j) \tag{38}$$

$$= \frac{1}{|\mathcal{A} \sqcup \mathcal{B}|} \sum_{z \in \mathcal{A} \sqcup \mathcal{B}} z \tag{39}$$

$$> \frac{1}{|\mathcal{A}|} \sum_{z \in \mathcal{A}} z \tag{40}$$

$$= \frac{1}{|\mathcal{E}|} \sum_{(v_i, v_j) \in \mathcal{E}} \text{sim}(v_i, v_j) \tag{41}$$

$$= \text{hom}(\mathcal{G}). \tag{42}$$

**Number of edges.** Since there are $|\mathcal{V}|$ nodes in total, then the total number of node pairs is $\binom{|\mathcal{V}|}{2}$. Recall that $\mathcal{E}^\dagger \setminus \mathcal{E}$ is the set of added edges. It follows that

$$|\mathcal{E}^\dagger| - |\mathcal{E}| = |\mathcal{E}^\dagger \setminus \mathcal{E}| \leq \binom{|\mathcal{V}|}{2} \tag{43}$$

$$= \frac{|\mathcal{V}|(|\mathcal{V}| - 1)}{2} = O(|\mathcal{V}|^2). \tag{44}$$

### D.4 PROOF OF OBSERVATION 2

Since $\text{sim}(v_i, v_j) = \|\boldsymbol{X}[v_i, :] \wedge \boldsymbol{X}[v_j, :]\|_\infty > 0$, then there exists a feature $k \in \mathcal{X}$ such that $\boldsymbol{X}[v_i, k] \wedge \boldsymbol{X}[v_j, k] > 0$. Since the feature matrix $\boldsymbol{X}$ is binary, then we must have

$$\boldsymbol{X}[v_i, k] = 1, \qquad \boldsymbol{X}[v_j, k] = 1. \tag{45}$$

This implies that $(v_i, x_k) \in \mathcal{E}^*$ and that $(v_j, x_k) \in \mathcal{E}^*$. Hence, there exists a length-2 path $v_i \to x_k \to v_j$ connecting graph nodes $v_i$ and $v_j$. Therefore, $v_i$ and $v_j$ are two-hop neighbors of each other.

### D.5 PROOF OF THEOREM 3

**Homophily.** Since the original graph $\mathcal{G}$ is homophilic, then there exists a pair of nodes, $v_i, v_j \in \mathcal{V}$ ($v_i \neq v_j$), such that $\text{sim}(v_i, v_j) = \|\boldsymbol{X}[v_i, :] \wedge \boldsymbol{X}[v_j, :]\|_\infty > 0$ but $(v_i, v_j) \notin \mathcal{E}$. Since $\text{sim}(v_i, v_j) = \|\boldsymbol{X}[v_i, :] \wedge \boldsymbol{X}[v_j, :]\|_\infty > 0$, then there exists a feature $k \in \mathcal{X}$ such that $\boldsymbol{X}[v_i, k] \wedge \boldsymbol{X}[v_j, k] > 0$. Since the feature matrix $\boldsymbol{X}$ is binary, then we must have

$$\boldsymbol{X}[v_i, k] = 1, \qquad \boldsymbol{X}[v_j, k] = 1. \tag{46}$$

This implies that $(v_i, x_k) \in \mathcal{E}^* \setminus \mathcal{E}$ and that $(v_j, x_k) \in \mathcal{E}^* \setminus \mathcal{E}$. Thus, $\mathcal{E}^* \setminus \mathcal{E}$ is nonempty.

Furthermore, for any feature node $x_k \in \mathcal{V}_\mathcal{X}$, since any feature edge $(v_i, x_k) \in \mathcal{E}_\mathcal{X}$ ensures $\boldsymbol{X}[v_i, k] = 1$, then we have

$$\boldsymbol{X}^*[x_k, k] = \frac{1}{|\mathcal{E}_\mathcal{X} \cap (\mathcal{V} \times \{x_k\})|} \sum_{v_i : (v_i, x_k) \in \mathcal{E}_\mathcal{X}} \boldsymbol{X}[v_i, k] \tag{47}$$

$$= \frac{1}{|\mathcal{E}_\mathcal{X} \cap (\mathcal{V} \times \{x_k\})|} \sum_{v_i : (v_i, x_k) \in \mathcal{E}} 1 \tag{48}$$

$$= \frac{1}{|\mathcal{E}_\mathcal{X} \cap (\mathcal{V} \times \{x_k\})|} \sum_{v_i : (v_i, x_k) \in \mathcal{E} \cap (\mathcal{V} \times \{x_k\})} 1 \tag{49}$$

$$= 1. \tag{50}$$

Finally, for any added feature edge $(v_i, x_k) \in \mathcal{E}^* \setminus \mathcal{E} = \mathcal{E}_\mathcal{X}$,

$$\text{sim}(v_i, x_k) = \|\boldsymbol{X}[v_i, :] \wedge \boldsymbol{X}[x_k, :]\|_\infty \tag{51}$$

$$= \max_{k' \in \mathcal{X}} |\boldsymbol{X}[v_i, k'] \wedge \boldsymbol{X}[x_k, k']| \tag{52}$$

$$\geq |\boldsymbol{X}[v_i, k] \wedge \boldsymbol{X}[x_k, k]| \tag{53}$$

$$= |1 \wedge 1| = 1. \tag{54}$$

Since $\text{hom}(\mathcal{G}) < 1$, then

$$\text{sim}(v_i, x_k) \geq 1 > \text{hom}(\mathcal{G}). \tag{55}$$

Therefore, by Lemma 4 with

$$\mathcal{A} := \{\!\!\{\text{sim}(v_i, v_j) : (v_i, v_j) \in \mathcal{E}\}\!\!\}, \tag{56}$$

$$\mathcal{B} := \{\!\!\{\text{sim}(v_i, x_k) : (v_i, x_k) \in \mathcal{E}_\mathcal{X}\}\!\!\}, \tag{57}$$

we have

$$\text{hom}(\mathcal{G}^*) = \frac{1}{|\mathcal{E}^*|} \sum_{(u,u') \in \mathcal{E}^*} \text{sim}(u, u') \tag{58}$$

$$= \frac{1}{|\mathcal{E} \sqcup \mathcal{E}_\mathcal{X}|} \sum_{(u,u') \in \mathcal{E} \sqcup \mathcal{E}_\mathcal{X}} \text{sim}(u, u') \tag{59}$$

$$= \frac{1}{|\mathcal{A} \sqcup \mathcal{B}|} \sum_{z \in \mathcal{A} \sqcup \mathcal{B}} z \tag{60}$$

$$> \frac{1}{|\mathcal{A}|} \sum_{z \in \mathcal{A}} z \tag{61}$$

$$= \frac{1}{|\mathcal{E}|} \sum_{(v_i, v_j) \in \mathcal{E}} \text{sim}(v_i, v_j) \tag{62}$$

$$= \text{hom}(\mathcal{G}). \tag{63}$$

**Number of nodes.** Since $|\mathcal{X}| \leq O(|\mathcal{V}|)$, then

$$|\mathcal{V}_\mathcal{X}| = |\mathcal{X}| \leq O(\mathcal{V}). \tag{64}$$

It follows that

$$|\mathcal{V}^*| = |\mathcal{V}| + |\mathcal{V}_\mathcal{X}| \tag{65}$$

$$\leq |\mathcal{V}| + O(|\mathcal{V}|) \tag{66}$$

$$= O(|\mathcal{V}|). \tag{67}$$

**Number of edges.** Since $\boldsymbol{X}$ is a binary matrix, then $\|\boldsymbol{X}\|_1 = \|\boldsymbol{X}\|_0 \leq O(|\mathcal{E}|)$. Hence,

$$|\mathcal{E}_\mathcal{X}| = \sum_{v_i \in \mathcal{V}} \sum_{x_k \in \mathcal{V}_\mathcal{X}} 1_{[(v_i, x_k) \in \mathcal{E}_\mathcal{X}]} \tag{68}$$

$$= \sum_{v_i \in \mathcal{V}} \sum_{k \in \mathcal{X}} 1_{[(v_i, x_k) \in \mathcal{E}_\mathcal{X}]} \tag{69}$$

$$= \sum_{v_i \in \mathcal{V}} \sum_{k \in \mathcal{X}} 1_{[\boldsymbol{X}[v_i, k] = 1]} \tag{70}$$

$$= \sum_{v_i \in \mathcal{V}} \sum_{k \in \mathcal{X}} \boldsymbol{X}[v_i, k] \tag{71}$$

$$= \sum_{v_i \in \mathcal{V}} \sum_{k \in \mathcal{X}} |\boldsymbol{X}[v_i, k]| \tag{72}$$

$$= \|\boldsymbol{X}\|_1 = \|\boldsymbol{X}\|_0 \leq O(|\mathcal{E}|). \tag{73}$$

It follows that

$$|\mathcal{E}^*| = |\mathcal{E}| + |\mathcal{E}_\mathcal{X}| \tag{74}$$

$$\leq |\mathcal{E}| + O(|\mathcal{E}|) \tag{75}$$

$$= O(|\mathcal{E}|). \tag{76}$$

# E  USE OF LARGE LANGUAGE MODELS

We made limited and controlled use of large language models (LLMs) solely for stylistic refinement and improving readability of the text. All scientific content, methodology, experiments, and conclusions were fully conceived and validated by the authors. The role of LLMs was purely editorial and does not constitute co-authorship.

