# OpenReview forum: "Graph homophily booster: Reimagining the role of discrete features in heterophilic graph learning"
_ICLR.cc/2026/Conference — ICLR 2026 Poster_

### Official Review · Reviewer_aJfv · 2025-10-29

**Soundness:** 2
**Presentation:** 3
**Contribution:** 2
**Rating:** 6
**Confidence:** 4

**Summary:**

This paper proposes GRAPHITE, a simple graph transformation that explicitly increases homophily before running any GNN: it adds a set of feature nodes (hubs) and connects each original node to the hubs matching its discrete features, so that nodes with similar features become two hops apart and message passing becomes homophilic.

**Strengths:**

The motivation is straightforwad and the presentation is easy to understand.

**Weaknesses:**

Some motivation and explanation need more solid justifications.

**Questions:**

1. It is found that higher homophily does not always indicate a better graph for GNNs, i.e. mid-homophily pitfall [1]. How does the motivation of your proposed homophily booster align with this conclusion?

2. More experiments on large scale datasets used in [2].

3. What is the motivation or intuition behind the definition of shortcut connection, which connect nodes that share at least one feature dimension?

4. Naive homophily booster (NHB) oversimplifies the distribution of features. As shown in [3], there exist both good and bad features for message passing, and if you connect two nodes when they share a bad feature, you will not obtain a graph with higher homophily.

5. To my understanding, the number of feature nodes is the same as the number of feature dimension. For the tasks they have very large feature dimensions, this will significantly increase the number of nodes in the graph. What is the computational complexity of your algorithm?

6. Will the connection to "hub” nodes cause over-squashing problem?

7. Missing baselines in your comparison, e.g. ACMGNN [4], FSGNN [5]


[1] When do graph neural networks help with node classification? investigating the homophily principle on node distinguishability. Advances in Neural Information Processing Systems. 2024 Feb 13;36.

[2] Finding global homophily in graph neural networks when meeting heterophily. In International Conference on Machine Learning, pp. 13242–13256. PMLR, 2022.

[3] Let Your Features Tell The Differences: Understanding Graph Convolution By Feature Splitting. InThe Thirteenth International Conference on Learning Representations.


[4] Revisiting heterophily for graph neural networks. Advances in neural information processing systems. 2022 Dec 6;35:1362-75.


[5] Simplifying approach to node classification in graph neural networks. Journal of Computational Science, 62, 101695.

---

> ### Author Response · Authors · 2025-11-24
> **Official Comment (Part 1/2)**
>
> Thank you so much for providing such an insightful review! We provide a point-to-point response as follows.
>
> > Q1: It is found that higher homophily does not always indicate a better graph for GNNs, i.e. mid-homophily pitfall [1]. How does the motivation of your proposed homophily booster align with this conclusion?
>
> Thanks for the insightful question. The analysis of mid-homophily pitfall [1] assumes that the graph is CSBM-H. Since our feature nodes make the graph structure quite different from CSBM-H, the relationship between performance and homophily might be different from the analysis in [1]. As observed in our experiments, our method improves the accuracy and homophily simultaneously, so the modified graph structure may have made a difference.
>
> A possible explanation from the Node Distinguishability (ND) perspective in [1] is as follows.
> - Regarding intra-class ND, nodes in the same class might share many common features, so including these feature nodes into their neighborhoods might reduce the intra-class ND, which is beneficial according to [1].
> - Regarding inter-class ND, nodes in different classes might have different combinations of features, so including these feature nodes into their neighborhoods might increase the inter-class ND, which is beneficial as well according to [1].
>
> > Q2: More experiments on large scale datasets used in [2].
>
> Thanks for the suggestion. Unfortunately, due to hardware constraints, we are unable to test too large graphs. To justify the efficiency, we provide a comparison of running times with and without our transformation, respectively. From the table below (sorted by graph size), we can see that **our graph transformation has only a small impact on running time** while significantly improves accuracy. Notably, **the computational overhead gets smaller on larger graphs** (e.g., Minesweeper), justifying the scalability of our method.
>
> |Method|Minesweeper|Actor|Squirrel-F|Chameleon-F|
> |:--|:-:|:-:|:-:|:-:|
> |No transformation|1.9min|1.5min|0.7min|0.5min|
> |With transformation|2.3min|2.0min|1.1min|0.7min|
>
> > Q3: What is the motivation or intuition behind the definition of shortcut connection, which connect nodes that share at least one feature dimension?
>
> Our main motivation is as follows. Firstly, if features are informative, then feature homophily should be highly correlated with label homophily. Thus, if we can improve feature homophily, then it is likely that label homophily will be improved as a result, which is good for node classification. To improve feature homophily, a natural idea is to add shortcuts between nodes with similar features. For discrete features, this means that nodes sharing more common features should have more shortcuts. The feature nodes in GRAPHITE naturally serve as such shortcuts. In particular, if two nodes have many common features, then they will be connected by many parallel shortcuts, facilitating the homophilic message passing between them.
>
> > Q4: Naive homophily booster (NHB) oversimplifies the distribution of features. As shown in [3], there exist both good and bad features for message passing, and if you connect two nodes when they share a bad feature, you will not obtain a graph with higher homophily.
>
> Thanks for the insightful question. We present NHB as a motivating extreme (Sec.3.1) and then propose GRAPHITE to make that idea effective and efficient. In GRAPHITE, two nodes sharing features are connected by multiple feature nodes, so as long as most of the features are good for message passing, then the adverse impact of bad features will be negligible.
>
> On the other hand, if most of the features are bad features for message passing, maybe we can apply the method in [3] to identify the good features first and only use the good features to construct feature nodes in GRAPHITE. We will discuss [3] in our paper.

---

> ### Author Response · Authors · 2025-11-24
> **Official Comment (Part 2/2)**
>
> > Q5: To my understanding, the number of feature nodes is the same as the number of feature dimension. For the tasks they have very large feature dimensions, this will significantly increase the number of nodes in the graph. What is the computational complexity of your algorithm?
>
> As implied by Eq(4), the space complexity of the feature edges is at most proportional to the space complexity of the node feature matrix X. If the feature dimensionality is high, then the node feature matrix X must be very sparse (or otherwise the storage cost of X (num nodes \* num features) would be prohibitive).
>
> In this regard, since the time complexity of our method is proportional to the space complexity of feature edges, the time complexity is comparable to the GNN (up to constant factors).
>
> > Q6: Will the connection to "hub” nodes cause over-squashing problem?
>
> Thanks for the insightful question. We agree that for each single feature node, over-squashing can happen, but this design is intended for the following two considerations:
> 1. For two nodes with many common features, they both connect to many feature nodes, so their message passing will go through many parallel paths, alleviating the over-squashing of messages between them. On the other hand, for two dissimilar nodes, the heterophilic message passing between them will be supressed due to over-squashing.
> 2. Thanks to over-squashing, each feature node has to "select" only information related to this feature and filter out other information. This encourages the representation diversity among feature nodes and thus potentially improves the "learned expressiveness" of the GNN.
>
> > Q7: Missing baselines in your comparison, e.g. ACMGNN [4], FSGNN [5]
>
> Thanks for suggesting the baselines. In the table below (on Actor), we can see that our simple method still outperforms [4,5]. We will add the complete results to Table 1.
>
> |Ours|ACMGNN[4]|FSGNN[5]|
> |:--|:-:|:-:|
> |**37.69**|36.63|35.67|

---

### Official Review · Reviewer_6ZKr · 2025-10-30

**Soundness:** 3
**Presentation:** 3
**Contribution:** 2
**Rating:** 6
**Confidence:** 4

**Summary:**

This paper proposes GRAPHITE (GRAph homoPHIly boosTEr), a novel framework that directly enhances graph homophily on heterophilic graphs through an explicit structural transformation. Unlike prior methods that rely on architectural modifications, GRAPHITE introduces “feature nodes” as intermediaries to indirectly connect original nodes sharing similar discrete features—significantly boosting homophily with only a modest increase in graph size. The authors provide theoretical guarantees for this homophily gain and demonstrate through extensive experiments that GRAPHITE substantially outperforms state-of-the-art methods on heterophilic benchmarks, while also improving the performance of standard homophilic GNNs when applied to the transformed graph.

**Strengths:**

1. Paradigm shift:  GRAPHITE directly enhances graph homophily through structural transformation, offering a fundamental and effective new perspective. Moreover, its design is simple and efficient: by using feature nodes as hubs, it avoids the O(|V|²) edge explosion and achieves provably improved homophily with only O(|V|) added nodes and O(|E|) added edges.

2. It elegantly decouples feature similarity from the original graph structure to enable semantics-aware message passing: by introducing feature nodes that explicitly connect nodes sharing discrete features, GRAPHITE preserves the original topology while adding semantic similarity-based propagation paths—allowing GNNs to naturally integrate structural and semantic homophilic signals without architectural changes.

3. Comprehensive and convincing experiments: GRAPHITE consistently outperforms more than 25 state-of-the-art methods across four heterophilic datasets and significantly boosts the performance of standard homophilic GNNs when applied to the transformed graph.

**Weaknesses:**

1. Although the authors list the hyperparameter search ranges and training configurations in Section 4.1 and Appendix B.3, the paper lacks sensitivity analysis of these parameters, which weakens the robustness argument of the method.

2. The homophily definition used in the theoretical analysis (based on feature intersection) is inconsistent with the experimental metrics (feature/adjusted homophily). Although the two are intuitively positively correlated, the paper lacks rigorous derivation or empirical validation—e.g., is the theoretical homophily strongly correlated with the experimental homophily metrics across all datasets?

3. Feature nodes only connect nodes that share the same discrete features, which may cause the GNN to aggregate neighbors based on "shared keywords" rather than "same-class labels." Can case studies or ablation experiments demonstrate that feature nodes indeed promote semantic or label-level homophily, rather than merely feature-level aggregation?

**Questions:**

See Weakness.

---

> ### Author Response · Authors · 2025-11-24
>
> Thank you so much for your recognition of our work! We provide a point-to-point response as follows.
>
> > W1: Although the authors list the hyperparameter search ranges and training configurations in Section 4.1 and Appendix B.3, the paper lacks sensitivity analysis of these parameters, which weakens the robustness argument of the method.
>
> Thanks for the suggestion. We agree that a senstivity analysis would strengthen this paper.
>
> As shown in the tables below (e.g. Minesweeper), we see that our method is not sensitive to these hyperparameters, and **our method consistently outperform the best baseline under various hyperparameters**. Hence, we believe hyperparameter tuning is not hard for our method.
>
> |Best Baseline|Ours, $\tau=0.1$|Ours, $\tau=0.5$|Ours, $\tau=1.0$|Ours, $\tau=1.5$|Ours, $\tau=2.0$|
> |:--|:-:|:-:|:-:|:-:|:-:|
> |91.60|93.48|94.29|94.78|94.66|94.18|
>
> |Best Baseline|Ours, $w_X=0.1$|Ours, $w_X=0.25$|Ours, $w_X=0.5$|Ours, $w_X=0.75$|Ours, $w_X=1.0$|
> |:--|:-:|:-:|:-:|:-:|:-:|
> |91.60|94.78|94.16|93.95|93.53|93.15|
>
> > W2: The homophily definition used in the theoretical analysis (based on feature intersection) is inconsistent with the experimental metrics (feature/adjusted homophily). Although the two are intuitively positively correlated, the paper lacks rigorous derivation or empirical validation—e.g., is the theoretical homophily strongly correlated with the experimental homophily metrics across all datasets?
>
> Thanks for the suggestion. We agree that adding an empirical validation would strengthen the theoretical claim of this paper. In fact, our **Table 2** shows that our method empirically improves the experimental metrics despite the slight difference between theoretical and experimental metrics. Furthermore, **we have also observed positive correlation** between our theoretical and experimental metrics. For instance, our theoretical homophily and the feature homophily have Pearson correlation 0.73 (averaged over datasets), which is a considerably high correlation. This further justifies the definition of our theoretical homophily.
>
> > W3: Feature nodes only connect nodes that share the same discrete features, which may cause the GNN to aggregate neighbors based on "shared keywords" rather than "same-class labels." Can case studies or ablation experiments demonstrate that feature nodes indeed promote semantic or label-level homophily, rather than merely feature-level aggregation?
>
> Thanks for the question. In fact, the right half of our Fig2 shows that feature nodes significantly improves label-level homophily (e.g., 28.67% on Actor), where "adjusted homophily" (Platonov et al., 2024) is a **label-level** homophily metric. From Fig2 we can see that our method improves both feature homophily and label-level homophily. Intuitively, this is because as long as features are informative for classification, then feature homophily should be positively correlated with label-level homophily.

---

### Official Review · Reviewer_3w7g · 2025-10-31

**Soundness:** 2
**Presentation:** 2
**Contribution:** 2
**Rating:** 2
**Confidence:** 4

**Summary:**

This paper introduces the GRAPHITE method, which improves the performance of Graph Neural Networks (GNNs) on heterophilic graphs by introducing feature nodes to increase the graph's homophily. Unlike existing approaches, GRAPHITE directly enhances homophily through graph transformation rather than architectural changes. The method is theoretically proven to be effective and is validated through extensive experiments, demonstrating significant improvements on heterophilic datasets. Results show that GRAPHITE outperforms state-of-the-art methods on heterophilic graphs while remaining competitive on homophilic graphs.

**Strengths:**

1)	Introduces a graph transformation method that effectively boosts homophily on heterophilic graphs.
2)	Both theoretical analysis and empirical experiments support the superior performance of GRAPHITE on heterophilic graphs.
3)	Demonstrates good compatibility and enhancement effects with existing GNN architectures.

**Weaknesses:**

1)	Limited Novelty in Core Idea: While the paper introduces the GRAPHITE method to boost homophily in heterophilic graphs, the fundamental concept of adding feature nodes or shortcut connections between similar nodes is not entirely novel. Many prior works have explored similar strategies for improving graph neural network (GNN) performance, such as feature augmentation or graph transformation methods. The core idea lacks significant innovation, which might reduce its contribution to the broader research community.

2)	Scalability Concerns: Although the paper claims that the proposed method is computationally efficient, it does not provide a detailed analysis of how GRAPHITE performs on very large-scale graphs. The addition of feature nodes and edges might still become a bottleneck in cases with millions of nodes, limiting the method's practical applicability for large datasets. More empirical evaluation on large graphs or an analysis of the method’s time complexity would be valuable.

3)	Narrow Focus on Discrete Features: The proposed method mainly focuses on graphs with discrete node features. However, many real-world applications (e.g., social networks, biological networks) involve continuous features. The paper does not address how GRAPHITE would perform in such scenarios or whether the approach could be generalized to handle graphs with continuous or mixed-type features. This limits the generalizability of the method to broader domains.

**Questions:**

1、The paper repeatedly claims to be the first to propose a graph transformation method for improving performance on heterophilic datasets. However, there are numerous prior works, such as those by [1] and [2], that have also modified graph structures by introducing virtual nodes to enhance performance on heterophilic graphs. Unfortunately, the paper does not adequately discuss or compare these related studies

[1]Dong Y, Dupty M H, Deng L, et al. Differentiable cluster graph neural network. ICLR 2025.
[2]Zhang A, Li P, Chen G. Steering graph neural networks with pinning control[J]. arXiv preprint arXiv:2303.01265, 2023.


2、The feature of a feature node is defined $V_{x}$ as the " we define its node feature as the average feature vector among the graph nodes $v_i$, that are connected to feature node $x_k$:" (Equation 5). However, the rationale behind this design has not been sufficiently justified. If the features of the graph nodes $v_i$, connected to the same feature node $x_k$ exhibit significant differences (for example, some nodes may share the feature k while having completely distinct other features), the average feature may not effectively represent the feature node and could even introduce noise. Has the author compared the effectiveness of other feature construction methods for characteristic nodes, such as majority voting or weighted averaging? Why has "mean" been selected as the optimal choice?

3、The ablation experiment did not analyze the sensitivity of hyperparameters such as "feature edge weights w_x" and "temperature parameter τ" to performance. Do the values of these parameters have significant dataset dependencies? If the dataset is changed, is the cost of hyperparameter tuning too high?

---

> ### Author Response · Authors · 2025-11-24
> **Official Comment (Part 1/2)**
>
> > W1: Limited Novelty in Core Idea: While the paper introduces the GRAPHITE method to boost homophily in heterophilic graphs, the fundamental concept of adding feature nodes or shortcut connections between similar nodes is not entirely novel. Many prior works have explored similar strategies for improving graph neural network (GNN) performance, such as feature augmentation or graph transformation methods. The core idea lacks significant innovation, which might reduce its contribution to the broader research community.
> >
> > Q1: The paper repeatedly claims to be the first to propose a graph transformation method for improving performance on heterophilic datasets. However, there are numerous prior works, such as those by [1] and [2], that have also modified graph structures by introducing virtual nodes to enhance performance on heterophilic graphs. Unfortunately, the paper does not adequately discuss or compare these related studies
>
> Firstly, we would like to clarify that our core contribution is **non-learning-based, guaranteed increase of homophily**, not just connecting similar nodes. Due to its non-learning-based nature, our graph transformation is effective **regardless of the GNN architecture**, as demonstrated in Table 3. Hence, we believe that our core contribution "non-learning-based, guaranteed increase of homophily" is still novel.
>
> We will discuss related works such as [1,2] in our paper. Notably, [1,2] cannot guarantee to increase homophily because they are learning-based, but our non-learning-based transform can guarantee homophily increase. (Unfortunately, since [1,2] are not open-source, we are unable to empirically compare with them.)
>
> > W2: Scalability Concerns: Although the paper claims that the proposed method is computationally efficient, it does not provide a detailed analysis of how GRAPHITE performs on very large-scale graphs. The addition of feature nodes and edges might still become a bottleneck in cases with millions of nodes, limiting the method's practical applicability for large datasets. More empirical evaluation on large graphs or an analysis of the method’s time complexity would be valuable.
>
> We would like to clarify that the number of added nodes (i.e., the number of features) is typically negligibly small compared with the number of nodes on large graphs. For example, ogbn-papers100M has over 1e8 nodes but only 128 features, so applying GRAPHITE to this graph would only add 128 nodes, which is negligibly small compared with the graph size.
>
> Regarding the feature edges, as implied by Eq(4), the storage cost of the feature edges is at most proportional to the storage cost of the node feature matrix X. In practice, discrete features are typically sparse, so the number of feature edges are even smaller.
>
> Unfortunately, due to hardware constraints, we are unable to test too large graphs. To justify the efficiency, we provide a comparison of running times with and without our transformation, respectively. From the table below (sorted by graph size), we can see that **our graph transformation has only a small impact on running time** while significantly improves accuracy. Notably, **the computational overhead gets smaller on larger graphs** (e.g., Minesweeper), justifying the scalability of our method.
>
> |Method|Minesweeper|Actor|Squirrel-F|Chameleon-F|
> |:--|:-:|:-:|:-:|:-:|
> |No transformation|1.9min|1.5min|0.7min|0.5min|
> |With transformation|2.3min|2.0min|1.1min|0.7min|
>
> > W3: Narrow Focus on Discrete Features: The proposed method mainly focuses on graphs with discrete node features. However, many real-world applications (e.g., social networks, biological networks) involve continuous features. The paper does not address how GRAPHITE would perform in such scenarios or whether the approach could be generalized to handle graphs with continuous or mixed-type features. This limits the generalizability of the method to broader domains.
>
> We agree GRAPHITE assumes discrete features (Sec 2.1) and have acknowledged continuous features as an interesting future work (Sec 6). This work focuses on discrete features because (i) discrete features are very common in real-world applications and (ii) even though in practice a graph may have both discrete and continuous features, we can still apply our method to its discrete features to boost the GNN performance.
>
> Possible extensions to continuous features include: (i) cluster the continuous features into a few clusters and create a feature node for each cluster; (ii) bin the continuous features into discrete buckets when creating feature nodes. As long as the continuous features are correlated with class labels, we may still be able to theoretically guarantee that this transformation improves homophily.

---

> ### Author Response · Authors · 2025-11-24
> **Official Comment (Part 2/2)**
>
> > Q2: The feature of a feature node $V_x$ is defined as "we define its node feature as the average feature vector among the graph nodes $v_i$, that are connected to feature node $x_k$:" (Equation 5). However, the rationale behind this design has not been sufficiently justified. If the features of the graph nodes $v_i$, connected to the same feature node $x_k$, exhibit significant differences (for example, some nodes may share the feature k while having completely distinct other features), the average feature may not effectively represent the feature node and could even introduce noise. Has the author compared the effectiveness of other feature construction methods for characteristic nodes, such as majority voting or weighted averaging? Why has "mean" been selected as the optimal choice?
>
> Thanks for the suggestion. We have actually tried a few variants before and finally chose averaging. In fact, **all aggregators have similar accuracy** (see table below), so we use averaging in our method due to its simplicity. We will add this ablation study to Sec 4.
>
> |Dataset|Averaging (Ours)|Majority Vote|Weighted Averaging|
> |:--|:-:|:-:|:-:|
> |Actor|37.69|37.59|37.13|
> |Squirrel-F|43.06|43.28|43.46|
> |Chameleon-F|45.08|45.64|44.36|
> |Minesweeper|94.78|94.75|94.56|
>
> > Q3: The ablation experiment did not analyze the sensitivity of hyperparameters such as "feature edge weights $w_x$" and "temperature parameter $\tau$" to performance. Do the values of these parameters have significant dataset dependencies? If the dataset is changed, is the cost of hyperparameter tuning too high?
>
> Thanks for the suggestion. We agree that a senstivity analysis would strengthen this paper.
>
> As shown in the tables below (e.g. Minesweeper), we see that our method is not sensitive to these hyperparameters, and **our method consistently outperform the best baseline under various hyperparameters**. Hence, we believe hyperparameter tuning is not hard for our method.
>
> |Best Baseline|Ours, $\tau=0.1$|Ours, $\tau=0.5$|Ours, $\tau=1.0$|Ours, $\tau=1.5$|Ours, $\tau=2.0$|
> |:--|:-:|:-:|:-:|:-:|:-:|
> |91.60|93.48|94.29|94.78|94.66|94.18|
>
> |Best baseline|Ours, $w_X=0.1$|Ours, $w_X=0.25$|Ours, $w_X=0.5$|Ours, $w_X=0.75$|Ours, $w_X=1.0$|
> |:--|:-:|:-:|:-:|:-:|:-:|
> |91.60|94.78|94.16|93.95|93.53|93.15|

---

> > ### Comment · Reviewer_3w7g · 2025-11-26
> >
> > Thank you for your response. It has addressed most of my concerns. However, the extensibility of the method, particularly its theoretical and empirical behavior when dealing with non-discrete features (e.g., Bag-of-Words vectors), remains unclear. In my assessment the paper falls around a borderline level—my intended score would have been a 5, but since that option is not available, I will assign a 6.

---

> > > ### Author Response · Authors · 2025-11-26
> > >
> > > Dear Reviewer 3w7g,
> > >
> > > Thank you so much for raising the score! We agree that extension to continuous features might be non-trivial, and it will be an interesting future work we would like to explore and investigate. We also deeply appreciate your insightful suggestions and comments, which we believe will considerably improve the paper. We will add the clarifications and the experiments to the paper.

---

### Official Review · Reviewer_3mQE · 2025-11-02

**Soundness:** 2
**Presentation:** 2
**Contribution:** 2
**Rating:** 4
**Confidence:** 5

**Summary:**

This paper introduces GRAPHITE, a novel graph transformation framework that addresses heterophily in GNNs by explicitly increasing graph homophily through the creation of feature nodes. Unlike existing methods that focus on architectural modifications, GRAPHITE directly transforms the graph structure by introducing feature nodes that serve as hubs to connect nodes with similar discrete features. The method provides theoretical guarantees for homophily improvement with minimal graph size increase and demonstrates strong empirical performance across multiple heterophilic benchmarks, outperforming state-of-the-art methods while maintaining competitive performance on homophilic graphs.

**Strengths:**

1. First work to explicitly transform graph structure to increase homophily, offering a fundamentally different approach from architectural GNN modifications.
2. Provides formal proofs showing GRAPHITE guarantees homophily improvement (Theorem 3) with only linear growth in graph size.
3. Consistently outperforms 25 baselines across 4 heterophilic datasets with improvements up to 5.35%, while maintaining competitive performance on homophilic graphs.

**Weaknesses:**

1. Method is specifically designed for discrete node features, limiting applicability to graphs with continuous features without discretization.
2. While theoretical complexity is linear, practical implementation with 8 GNN layers and 512 hidden dimensions may be computationally expensive for very large graphs.
3. Feature node representations (Equation 5) use simple averaging, potentially overlooking more sophisticated feature aggregation strategies.
4. No exploration of how the transformation affects graph properties like diameter, clustering coefficients, or other structural metrics beyond homophily.

**Questions:**

1. How can GRAPHITE be extended to handle continuous node features while maintaining theoretical guarantees?

2. What is the impact of the transformation on graph structural properties beyond homophily, and could certain properties be adversely affected?

3. Have the authors considered alternative feature node representations beyond averaging, such as learned embeddings or attention-weighted aggregations?

4. How does the method scale to extremely large graphs in practice, considering the increased node count from feature node addition?

---

> ### Author Response · Authors · 2025-11-24
>
> > W1: Method is specifically designed for discrete node features, limiting applicability to graphs with continuous features without discretization.
> >
> > Q1: How can GRAPHITE be extended to handle continuous node features while maintaining theoretical guarantees?
>
> We agree GRAPHITE assumes discrete features (Sec.2.1) and have acknowledged continuous features as an interesting future work (Sec 6). This work focuses on discrete features because (i) discrete features are very common in real-world applications and (ii) even though in practice a graph may have both discrete and continuous features, we can still apply our method to its discrete features to boost the GNN performance.
>
> Possible extensions to continuous features include: (i) cluster the continuous features into a few clusters and create a feature node for each cluster; (ii) bin the continuous features into discrete buckets when creating feature nodes. As long as the continuous features are correlated with class labels, we may still be able to theoretically guarantee that this transformation improves homophily.
>
> > W2: While theoretical complexity is linear, practical implementation with 8 GNN layers and 512 hidden dimensions may be computationally expensive for very large graphs.
> >
> > Q4: How does the method scale to extremely large graphs in practice, considering the increased node count from feature node addition?
>
> As shown in **Table 3**, even for very small GNNs (GCN, GAT, GraphSAGE, JKNet, GIN), adding our GRAPHITE to them still achieves significant improvement (for example, 30.21 to 34.83 for GCN on Actor). This demonstrates that our improvement is **regardless of the GNN size**, so we believe that our method is still applicable to very large graphs using such small GNNs.
>
> Regarding the node count, note that the number of added nodes (i.e., the number of features) is typically negligibly small compared with the number of nodes on large graphs.
>
> For example, ogbn-papers100M has over 1e8 nodes but only 128 features, so applying GRAPHITE to this graph would only add 128 nodes, which is negligibly small compared with the graph size.
>
> > W3: Feature node representations (Equation 5) use simple averaging, potentially overlooking more sophisticated feature aggregation strategies.
> >
> > Q3: Have the authors considered alternative feature node representations beyond averaging, such as learned embeddings or attention-weighted aggregations?
>
> Thanks for the suggestion. We have actually tried a few variants before and finally chose averaging. In fact, **all aggregators have similar accuracy** (see table below) while **averaging is more efficient** than learned embeddings and attention-weighted aggregation. Therefore, we use averaging in our method due to its efficiency. We will add this ablation study to Sec 4.
>
> |Dataset|Averaging (Ours)|Learned Embeds|Attention Aggr|
> |:--|:-:|:-:|:-:|
> |Actor|37.69|37.46|37.13|
> |Squirrel-F|43.06|43.53|43.46|
> |Chameleon-F|45.08|44.80|44.36|
> |Minesweeper|94.78|94.47|94.56|
>
> > W4: No exploration of how the transformation affects graph properties like diameter, clustering coefficients, or other structural metrics beyond homophily.
> >
> > Q2: What is the impact of the transformation on graph structural properties beyond homophily, and could certain properties be adversely affected?
>
> Firstly, we would like to clarify that **our goal is to improve node classifcation rather than preserve graph properties**. Considering the significant improvement from our graph transformations, we believe that our impact on graph properties is **beneficial** for node classification rather than adverse.
>
> For instance, as discussed in Observation 2 (page 5), our transformation ensures that any two nodes sharing a common feature will have distance at most two, so it potentially reduces the diamater of the graph. We believe that the reduced distances between similar nodes **facilitate homophilic message passing** and contribute to the significant improvement.

---

> > ### Comment · Reviewer_3mQE · 2025-11-25
> >
> > I appreciate the authors’ detailed response. I encourage the authors to include the relevant clarifications in the final version. I have decided to increase my score.

---

> ### Author Response · Authors · 2025-11-26
>
> Dear Reviewer 3mQE,
>
> Thank you so much for raising the score! We will include the clarifications and the results in the final version of the paper. We sincerely appreciate your constructive feedback, which we believe will considerably strengthen this paper.

---

### Meta-Review · Area_Chair_YeLT · 2026-01-01

**Summary:**

The reviewers generally agreed that this paper makes a clear and interesting contribution by introducing a simple yet effective graph transformation to directly improve homophily on heterophilic graphs. The initial concerns from the reviewers focused on scalability, sensitivity to hyperparameters, justification of feature-node design, missing baselines, and whether the method improves label-level homophily rather than only feature similarity. These concerns were largely addressed in the rebuttal. While some limitations remain, the reviewers’ major technical concerns were resolved, leading to improved scores and overall positive assessments.

**Reviewer Concerns:**

Reviewer Concerns Addressed by the Rebuttal

1. Multiple reviewers questioned whether GRAPHITE would be computationally expensive due to added feature nodes and edges. The authors provided both theoretical complexity analysis and empirical runtime comparisons, showing that the number of added nodes is bounded by the number of feature dimensions and is negligible for large graphs.

2. The reviewers asked whether feature edge weights and temperature parameters require careful tuning. The authors added sensitivity analysis across a wide range of hyperparameters and demonstrated that performance is stable and consistently better than baselines.

3. The reviewers pointed out missing comparisons with methods such as ACMGNN and FSGNN. The authors added these baselines and showed that GRAPHITE remains competitive or superior.

Outstanding Reviewer Concerns.

1. While the authors provided convincing complexity analysis and runtime comparisons, several reviewers explicitly requested experiments on much larger graphs (e.g., those used in prior large-scale heterophily studies). Due to hardware constraints, these experiments were not conducted. This leaves some uncertainty about end-to-end scalability in extremely large real-world settings.

2. Although well acknowledged and reasonably justified, the method is currently limited to discrete features. Extensions to continuous or mixed-type features remain future work, and no empirical validation is provided for such settings.

Most substantive concerns were convincingly addressed in the rebuttal through added analysis, experiments, and clarifications.

**Reviewer Scores:**

Reviewer 3mQE: This reviewer initially rated the paper to 4 due to scalability and design concerns. After detailed rebuttal and added analysis, the reviewer explicitly increased the score to 6.

Reviewer 3w7g: This reviewer raised concerns about novelty, discrete-feature restriction, and scalability. These were largely addressed during the rebuttal.  The reviewer indicated the paper falls at a borderline-to-positive level. They would like to assign a 5 if there were such a rating. Ultimately, they increase their score to 6.

Reviewer 6ZKr:  This reviewer originally gave generally positive score (6), with remaining minor concerns about robustness and definitions. The score would likely remain 6 after rebuttal.

Reviewer aJfv: This reviewer mentioned some conceptual and scalability questions but acknowledged the clarity and effectiveness of the method. The reviewer rated the paper at 6, and would likely remain this score.

---

### Decision · Program_Chairs · 2026-01-26

Accept (Poster)